

# Multiresponse modeling of an unsaturated zone isotope tracer experiment at the Landscape Evolution Observatory

Carlotta Scudeler[1,2], Luke Pangle[3], Damiano Pasetto[4], Guo-Yue Niu[5,6], Till Volkmann[6], Claudio Paniconi[1], Mario Putti[2], and Peter Troch[5,6]

[1]Institut National de la Recherche Scientifique, Centre Eau Terre Environnement (INRS-ETE), Université du Québec, Quebec City, Canada
[2]Department of Mathematics, University of Padova, Padova, Italy
[3]Department of Geosciences, Georgia State University, Atlanta, USA
[4]Laboratory of Ecohydrology, École Polytechnique Fédérale de Lausanne, Lausanne, Switzerland
[5]Biosphere 2 - Earth Science, University of Arizona, Tucson, USA
[6]Department of Hydrology and Water Resources, University of Arizona, Tucson, USA

*Correspondence to:* C. Scudeler (carlotta.scudeler@ete.inrs.ca)

**Abstract.** This paper explores the challenges of model parameterization and process representation when simulating multiple hydrologic responses from a highly controlled unsaturated flow and transport experiment with a physically-based model. The experiment, conducted at the Landscape Evolution Observatory (LEO), involved alternate injections of water and deuterium-enriched water into an initially very dry hillslope. The multivariate observations included point measures of water content and tracer concentration in the soil, total storage within the hillslope, and integrated fluxes of water and solute through the seepage face. The simulations were performed with a three-dimensional finite element model that solves the Richards and advection-dispersion equations. Integrated flow, integrated transport, distributed flow, and distributed transport responses were successively analyzed, with parameterization choices at each step supported by standard model performance metrics. In the first steps of our analysis, where seepage face flow, water storage, and average concentration at the seepage face were the target responses, an adequate match between measured and simulated variables was obtained using a simple parameterization consistent with that from a prior flow-only experiment at LEO. When passing to the distributed responses, it was necessary to introduce heterogeneity to additional soil hydraulic parameters to obtain an adequate match for the point-scale flow response. Augmented heterogeneity also improved the match against point measures of tracer concentration, although model performance here was considerably poorer. This suggests that still greater complexity is needed in the model parameterization, or that there may be gaps in process representation for simulating solute transport phenomena in very dry soils.

## 1 Introduction

Simulation models of water and solute interaction and migration through complex geologic media are essential tools for addressing fundamental and practical problems, ranging from basic scientific understanding of critical zone processes (Brooks et al., 2015) to improving the management of our freshwater resources (Gorelick and Zheng, 2015). Physically based distributed numerical models require a careful definition of spatially variable parameters and time variable boundary conditions, and can



produce information for numerous response variables at different levels of spatio-temporal aggregation. It is increasingly acknowledged that proper implementation and verification of these models, in terms of both process representation and parameter identification, requires detailed, multiresponse field or laboratory data, in contrast to traditional model evaluation based on a single, integrated response variable such as total discharge (Paniconi and Putti, 2015). However, multiobjective parameter es-

timation for nonlinear or coupled models with a high number of degrees of freedom is very challenging (Anderman and Hill, 1999; Keating et al., 2010), since classical techniques developed for simpler hydrological models (e.g., Gupta et al., 1998; Fenicia et al., 2007) are not readily extendable, in terms of robustness and efficiency, to more complex models. An important example of this complexity arises in the modeling of mass transport phenomena in unsaturated soils (e.g., Ghanbarian-Alavijeh et al., 2012; Russo et al., 2014).

While many hydrologic model assessment studies have reported good agreement between simulated and observed data when performance is measured against a single response variable, there are comparatively few studies that have made use of observation data from multiple response variables. Brunner et al. (2012), for instance, examined the performance of a one-dimensional (1D) unsaturated zone flow model when water table measurements were supplemented by evapotranspiration and soil moisture observations. Sprenger et al. (2015) assessed the performance of three inverse modeling strategies based on the

use of soil moisture and pore water isotope concentration data for a 1D unsaturated flow and transport model. Kampf and Burges (2007) obtained encouraging results for a 2D Richards equation flow model using integrated (subsurface outflow) and internal (piezometric water level and volumetric water content) measurements from a hillslope-scale experiment. Kumar et al. (2013) used multiple discharge measurements to calibrate and apply a distributed hydrologic model to 45 subcatchments of a river basin in Germany. Investigations based on hypothetical experiments are more common. Mishra and Parker (1989), for

example, obtained smaller errors for simultaneous estimation of flow and transport parameters than for sequential estimation based on synthetically-generated observations of water content, pressure head, and concentration.

In this study we perform a modeling analysis of the experimental data collected from an intensively-measured hillslope at the Landscape Evolution Observatory (LEO) of the Biosphere 2 facility (Hopp et al., 2009). The simulations were conducted with the CATHY (CATchment HYdrology) model (Camporese et al., 2010; Weill et al., 2011), a physics-based numerical

code that solves the 3D Richards and advection-dispersion equations and includes coupling with surface routing equations. The availability of extensive observational datasets from detailed multidisciplinary experiments (recent examples in addition to LEO include the TERENO network of experimental catchments (Zacharias et al., 2011) and the Chicken Creek artificial catchment  (Hofer et al., 2012)) can contribute vitally to testing and improving the current generation of integrated (surface-subsurface) hydrological models (Sebben et al., 2013; Maxwell et al., 2014).

Two experiments have been conducted to date at LEO, a rainfall and drainage test in February 2013 (Gevaert et al., 2014; Niu et al., 2014), which featured both subsurface and overland flow, and an isotope tracer test in April 2013  (Pangle et al., 2015), run under drier soil conditions and with reduced rainfall rates to avoid occurrence of surface runoff. Using both integrated (load cell and seepage face) and distributed (point-scale soil moisture and concentration) data collected during the tracer experiment, the objective of this study is to explore the challenges of multiresponse performance assessment for a 3D variably

saturated flow and solute transport model. In a first step we consider only integrated flow responses, and the CATHY model





is initially parameterized according to analyses of the February 2013 experiment. As integrated transport and point-scale flow and transport observations are progressively introduced in the analysis, the impact of different configurations (spatially uniform versus spatially variable parameters, treatment of initial and boundary conditions, etc) on the model's ability to capture the expanding and increasingly detailed response variables is examined.

## 2    Study site: Biosphere 2 Landscape Evolution Observatory

LEO is a large-scale community-oriented infrastructure managed by the University of Arizona at the Biosphere 2, Oracle, U.S.A. (Hopp et al., 2009; Huxman et al., 2009; Pangle et al., 2015). It consists of three identical convergent artificial landscapes (or hillslopes) constructed with the aim of advancing our predictive understanding of the coupled physical, chemical, biological, and geological processes at Earth's surface in changing climates. For the first years of LEO operation, vegetation is not present and the research is focused on the characterization of the hydrological response of the hillslopes in terms of water transit times, generation of seepage and overland flow, internal dynamics of soil moisture, and evaporation. The three hillslopes are of $10^o$ average slope and are 30 m long and 11 m wide. The landscapes are filled with 1 m of basaltic tephra ground to homogeneous loamy sand, chosen mainly for its primary elemental composition that includes critical nutrients for plant growth. The three landscapes are housed in a 2000 $m^2$ environmentally controlled facility. Each landscape contains a sensor and sampler network capable of resolving meter-scale lateral heterogeneity and submeter-scale vertical heterogeneity in water, energy, and carbon states and fluxes. The density of sensors and the frequency at which they can be polled allows for a monitoring intensity that is impossible to achieve in natural field settings. Additionally, each landscape has 10 load cells embedded into the structure that allow measurement of changes in total system mass and an engineered rain system that allows application of precipitation at rates between 2 and 40 mm/h. Tracers can be introduced into the system via the rainfall simulator at a constant or time-varying rate. The embedded soil water solution and soil gas samplers facilitate the use of these tracers to study water and solute movement within the hillslopes at a very dense spatial scale.

## 3    Methodology

### 3.1    Isotope tracer experiment

The first tracer experiment performed at the LEO-1 hillslope began at 9:30 am on April 13, 2013. The experiment consisted of three irrigation events that were applied over 10 days (Fig. 1). During each event the irrigation was applied at a rate of 12 mm/h for durations respectively of 5.5 h, 6 h, and 5.25 h. Irrigation was interrupted for 2.75 h during the third event (1.25 h from the start) due to necessary equipment maintenance, then restarted. During the second event deuterium-enriched water was introduced into the irrigation system. The enriched water had a hydrogen isotopic composition (expressed using the delta-notation as $\delta^2 H$) of approximately 0‰, which corresponds to an enrichment of approximately 60‰ compared to typical (non-enriched) irrigation source water.



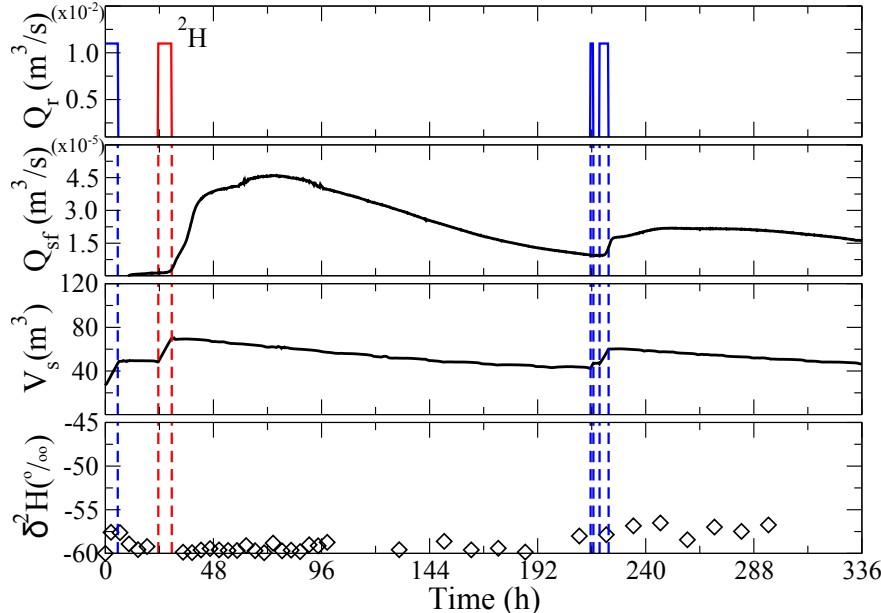

**Figure 1.** Hydrological response to the tracer experiment at the LEO-1 hillslope. From top: measured rain input pulses $Q_r$ (the red pulse is deuterium-enriched); seepage face flow $Q_{sf}$; total water storage $V_s$; and mean $\delta^2 H$ values at the seepage face. Time 0 corresponds to 9:30 am, 13 April 2013. The vertical dashed lines indicate the timing of the three pulses of rain (red when the water is deuterium-enriched and blue when it is not).

The initial conditions of the system were very dry. The estimated total initial volume of water was about 26 m³ (the total water storage capacity of the hillslope is approximately 130 m³). All the rain water applied infiltrated into the soil and generated seepage face outflow that started after 5 h. Two outflow peaks were observed: the first one after the second pulse of rain, with a peak of $4.5 \times 10^{-5}$ m³/s, and the second one after the final pulse, with a peak of $2.1 \times 10^{-5}$ m³/s. Temporal changes in total

soil water storage were monitored via the load cell measurements, flow from the seepage face boundary was measured with electronic flow meters and tipping bucket gauges, and matric potential and water content were measured at 496 locations with, respectively, MPS-2 and 5TM Decagon sensors installed at depths 5 cm, 20 cm, 50 cm, and 85 cm from the landscape surface. Cumulative fluxes and instantaneous state variables were recorded at 15-min intervals. The estimated evaporation rate derived based on water balance calculations was, on average, $1.9 \times 10^{-5}$ m³/s (5.0 mm/d) between rain pulses and $1.5 \times 10^{-5}$ m³/s

(3.9 mm/d) after the third rain pulse.

The movement of the deuterium-enriched water within and out of the landscape was monitored through manual sampling and subsequent laboratory analysis. Prenart quartz water sampling devices were used to extract soil water samples periodically throughout the experiment. Data reported in this manuscript include samples collected at 5, 20, 50, and 85 cm depth from surface at the four locations shown in Fig. 2. Flow from the seepage face boundary was collected with a custom autosampler

(sampling cylinders of 5 cm length and 3 cm circumference). The deuterium concentration within all water samples was



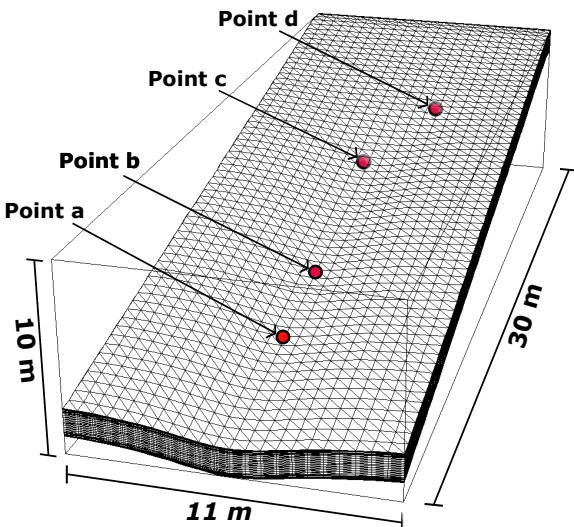

**Figure 2.** 3D numerical grid for the LEO landscape. Points a, b, c, and d are the locations where samples were extracted during the experiment for subsequent laboratory analysis.

measured via laser spectroscopy (LGR LWIA Model DLT-100) at the University of Arizona. Analytical precision was better than 0.5‰ for $\delta^2 H$. All isotopic data are expressed relative to the international reference VSMOW or VSMOW-SLAP scale.

## 3.2 Hydrological model

The CATHY (CATchment HYdrology) model (Camporese et al., 2010) used to simulate the isotope tracer experiment has
been previously implemented for LEO to study coupled surface and subsurface flow (Niu et al., 2014) and sensor performance (Pasetto et al., 2015). The description here will thus be limited to aspects pertaining particularly to the implementation for LEO of the solute transport component of the model. The numerical solver for the advection-dispersion transport equation is described in detail in Putti and Paniconi (1995), and, like the flow solver, is based on a three-dimensional finite element discretization in space and a weighted finite difference discretization in time. The velocity field and nodal saturation values
computed by the flow solver are passed as input at given time steps to the transport solver. The governing equations for the flow and transport solvers are:

$$S_w S_s \frac{\partial \psi}{\partial t} + n \frac{\partial S_w}{\partial t} = \nabla \cdot [K_r(\psi) K_s (\nabla \psi + \eta_z)] + q \tag{1}$$

$$\frac{\partial (n S_w c)}{\partial t} = \nabla \cdot (D \nabla c) - \nabla \cdot (vc) \tag{2}$$

where $S_w = \theta / \theta_s$ is the water saturation $[-]$, $\theta$ is the volumetric moisture content $[-]$, $\theta_s$ is the saturated moisture content $[-]$
(generally equal to the porosity $n$ $[-]$), $S_s$ is the aquifer specific storage coefficient $[1/L]$, $\psi$ is the pressure head $[L]$, $t$ is the





time $[T]$, $\nabla$ is the gradient operator $[1/L]$, $K_r(\psi)$ is the relative hydraulic conductivity function $[-]$, $K_s$ is the hydraulic conductivity tensor $[L/T]$ (considered to be diagonal, with $k_s$ the saturated hydraulic conductivity parameter for the isotropic case and $k_v$ and $k_h$, respectively, the vertical and horizontal hydraulic conductivity parameters for the anisotropic case), $\eta_z=(0,0,1)^T$, $z$ is the vertical coordinate directed upward $[L]$, $q$ is a source (when positive) or sink (when negative) term $[1/T]$, $c$ is the solute concentration $[M/L^3]$, $D$ is the dispersion tensor $[L^2/T]$, and $v = (v_1, v_2, v_3)^T$ is the Darcy velocity vector. The velocity vector is obtained from the flow equation as $v = -K_r K_s (\nabla \psi + \eta_z)$ while the dispersion tensor can be expressed as:

$$D_{ij} = nS_w \tilde{D}_{ij} = \alpha_t |v| \delta_{ij} + (\alpha_l - \alpha_t) \frac{v_i v_j}{|v|} + nS_w D_o \tau \delta_{ij} \quad i,j = 1,2,3 \tag{3}$$

where $|v| = \sqrt{v_1^2 + v_2^2 + v_3^2}$, $\alpha_l$ is the longitudinal dispersivity $[L]$, $\alpha_t$ is the transverse dispersivity $[L]$, $\delta_{ij}$ is the Kronecker delta $[-]$, $D_o$ is the molecular diffusion coefficient $[L^2/T]$, and $\tau$ is the tortuosity (we assume $\tau$=1) $[-]$. The evaluation of integrals arising in finite element discretization of the dispersion fluxes is performed using a rotated reference system spanned by the unit vectors $(x_1, x_2, x_3)$ that are aligned with the principal directions of anisotropy of $D$, whereby $x_1 = v/|v|$. Within this reference system, $D$ becomes diagonal, with the three components defined as:

$$D_{11} = \alpha_l |v| + nS_w D_o \tau \tag{4}$$

$$D_{22} = D_{33} = \alpha_t |v| + nS_w D_o \tau \tag{5}$$

The soil moisture–pressure head and relative conductivity–pressure head dependencies are described by the van Genuchten (1985) relationship:

$$S_w(\psi) = (1 - S_{wr}) S_e(\psi) + S_{wr} \tag{6}$$

$$K_r(\psi) = S_e^{0.5} \left[ 1 - (1 - S_e^{\frac{1}{m}})^m \right]^2 \tag{7}$$

$$S_e = \left[ 1 + \left( \frac{|\psi|}{|\psi_{sat}|} \right)^{n_{VG}} \right]^{\frac{1}{n_{VG}} - 1} \tag{8}$$

where $S_{wr}$ is the residual water saturation $[-]$, $S_e = (S_w - S_{wr})/(1 - S_{wr})$ is the effective saturation $[-]$, $n_{VG}$ is a fitting parameter ranging between 1.25 and 6 $[-]$, and $\psi_{sat}$ is related to the air entry suction $[L]$.

The transport equation (2) is solved in its conservative form, i.e., without applying the chain rule to the advective and storage terms. Using Euler time stepping, the resulting discretized system is:

$$([A+B]^{k+1} + \frac{1}{\Delta t_k} M^{k+1}) \hat{c}^{k+1} = \frac{1}{\Delta t_k} M^k \hat{c}^k - b^{t,k+1} \tag{9}$$





where $k$ is the time counter, $\hat{c}$ is the vector of the numerical approximation of $c$ at each node of the grid, and the coefficients of the, respectively, dispersion, advection, and mass matrices are:

$$a_{ij} = \int_{\Omega} D\nabla\phi_i\nabla\phi_j d\Omega \tag{10}$$

$$b_{ij} = \int_{\Omega} \nabla(v\phi_j)\phi_i d\Omega \tag{11}$$

$$m_{ij} = \int_{\Omega} nS_w\phi_i\phi_j d\Omega \tag{12}$$

where $i,j = 1,..,N$ with $N$ the number of nodes, $\Omega$ is the discretized domain, and $\phi$ are the basis functions of the Galerkin finite element scheme. The boundary condition vector for the discretized transport equation is:

$$b_i^t = \int_{\Gamma^t} (-D\nabla c)\cdot\nu\phi_i d\Gamma^t = \int_{\Gamma^t} q_n^t \phi_i d\Gamma^t \tag{13}$$

where $\Gamma^t$ is the boundary of the domain $\Omega$, $q_n^t$ $[M/(L^2T)]$ is the Neumann (dispersive) flux, and $\nu$ is the outward normal vector to the boundary. Cauchy, or mixed, boundary conditions can be easily implemented as variations of Eq. (13), involving an additional term in the system matrix implementing the advective component of the Cauchy condition.

### 3.3 Model setup for the LEO tracer experiment

We discretized the 30 m x 11 m x 1 m LEO hillslope into 60 x 22 grid cells in the lateral direction and 30 layers in the vertical direction (Fig. 2). The resulting surface mesh consists of 1403 nodes and 2640 triangular elements. The surface mesh was projected vertically to form a 3D tetrahedral mesh with parallel layers of varying thickness, with the thinnest layers assigned to the surface and bottom layers. This allows the numerical model to accurately capture infiltration/evaporation processes at the surface and the formation of base flow at the bottom of the domain. From top to bottom the thickness of the 30 layers is: 0.01 m for the first five layers, 0.025 m from layer 6 to layer 9, 0.05 m for layer 10, 0.06 m from layer 11 to layer 20, 0.05 m for layer 21, 0.025 m from layer 22 to layer 25, and 0.01 m from layer 26 to layer 30.

Measurements showed that the average $\delta^2 H$ of the irrigation source water at LEO was -60‰. For the transport model, we used a normalized concentration defined as:

$$c = \frac{\delta^2 H_{ref} - \delta^2 H}{\delta^2 H_{ref}} \tag{14}$$

where $\delta^2 H_{ref}$=-60‰ and $\delta^2 H$ is the actual value. Thus the initial conditions, as well as the concentrations of the first and third pulses, were $c=0$, while the second pulse had an imposed concentration of $c=1$.

A careful treatment of boundary conditions was essential to modeling the isotope tracer experiment, in particular at the land surface where three different cases needed to be considered: 1) Rain with $^2H$-enriched water (second pulse), handled as a Neumann prescribed flux condition for flow ($q_n^f = -K_s K_r(\psi)(\nabla\psi + \eta_z)\cdot\nu = v\cdot\nu$ with $q_n^f$=-12 mm/h) and a Cauchy prescribed advective flux condition for transport ($q_c^t = (vc - D\nabla c)\cdot\nu = v\cdot\nu c^*$ with $c^* = 1$); 2) Rain with no $^2H$-enriched water (first and





third pulses), handled with the same Neumann condition as case 1 for flow ($q_n^f = -12$ mm/h) and a zero Cauchy prescribed total flux condition for transport ($q_c^t = (vc - D\nabla c) \cdot \nu = 0$ with the concentration values at the surface nodes computed by the model); 3) Evaporation (between rain pulses and after the third pulse), handled with the same Neumann condition as case 1 for flow but with $q_n^f$=5.0 mm/d between the rain pulses and $q_n^f$=3.9 mm/d after the third pulse, and a zero Neumann prescribed dispersive flux condition for transport ($q_n^t = -D\nabla c \cdot \nu = 0$ with the concentration values at the surface nodes computed by the model). With the zero dispersive flux condition of case 3, all the isotopic mass in solution with the evaporating water leaves the domain by advection.

In addition to this "base case" treatment of rainfall and evaporation, we also introduced some variations on the surface boundary conditions. For rainfall (cases 1 and 2 above), we tested both uniform and variable spatial distributions. For the latter, a rainfall pattern with slightly higher rates towards the center of the landscape was used, as indicated by measurements taken during testing of the engineered rain system. This pattern was generated in such a way that the mean rainfall rate and the total volume of water injected were preserved. For evaporation, since there were no measurements of soil evaporation isotopic composition at the LEO landscape, we tested two other hypotheses — that none or only a portion (fractionation) of the isotope tracer evaporated — in addition to the zero dispersive flux condition of case 3.

To prevent isotope tracer from leaving the system through the landscape surface, we treated the evaporation as a sink term in the flow model, distributed exponentially from the surface to a depth of 38 cm, rather than as a Neumann boundary condition. The reasoning here is that evaporation involves not just the surface but also deeper soil layers. In generating the sink term, we ensured that the total volume of water evaporated was the same as in the Neumann boundary condition treatment. The sink term function $q$ in Eq. (1) applied to each layer $i$ ($i$=1,...,13 for a total depth of 38 cm) is:

$$q_i = \frac{F_{ev}}{\sum_{i=1}^{13}(e^{-\lambda z_i}\Delta z_i)}e^{-\lambda z_i} \tag{15}$$

where $q_i$ is applied to each tetrahedron of layer $i$, $\lambda$ [$1/L$] is a parameter set to 1 m$^{-1}$ in this case, $z_i$ is the depth from surface of layer $i$, $\Delta z_i$ is the thickness of layer $i$, and $F_{ev}$ [$L/T$] is the homogeneous evaporative flux used in the Neumann boundary condition case (with rates -5.7$\times10^{-8}$ m/s between rain pulses and -3.4$\times10^{-8}$ m/s after the third pulse). The applied sink fluxes are reported in Table 1. To ensure that all the tracer mass stays in the system, for the transport model we added a correction term $f_c$ in Eq. (2) and set it equal to $-qc$. In this way we inject back into the system the same amount of tracer mass that has exited with the sink term $q$.

Most land surface hydrological models still neglect fractionation, even though it can significantly influence the mass exchange at the land surface and the concentration profiles in the soil. Barnes and Allison (1988) examined isotope transport phenomena under both saturated and unsaturated conditions. In the latter case they experimentally observed that at steady state the maximum concentration of the heavier isotope species (e.g., $^2H$) occurs a short distance below the surface and decreases rapidly beyond that depth. The resulting profile can be explained as the result of vapor diffusion and isotopic exchange dominating the zone above the drying front and the balance between capillary and diffusive liquid water transport below the drying front (Craig and Gordon, 1965; Clark and Fritz, 1997; Horita et al., 2008). Alternative conceptualizations of the fractionation





**Table 1.** Sink term applied to the top 13 layers of the LEO grid. $q_{1i}$ is the sink term between the rain pulses 1,2, and 3 while $q_{2i}$ is the sink term after the third pulse. $z_i$ is the depth of the $i^{th}$ layer and $\Delta z_i$ is its thickness.

| Layer | $z_i$ (m) | $\Delta z_i$ (m) | $q_{1i}$ (1/s) | $q_{2i}$ (1/s) |
|---|---|---|---|---|
| 1 | 0.005 | 0.01 | $-1.79 \times 10^{-7}$ | $-1.07 \times 10^{-7}$ |
| 2 | 0.015 | 0.01 | $-1.77 \times 10^{-7}$ | $-1.06 \times 10^{-7}$ |
| 3 | 0.025 | 0.01 | $-1.75 \times 10^{-7}$ | $-1.05 \times 10^{-7}$ |
| 4 | 0.035 | 0.01 | $-1.74 \times 10^{-7}$ | $-1.04 \times 10^{-7}$ |
| 5 | 0.045 | 0.01 | $-1.72 \times 10^{-7}$ | $-1.03 \times 10^{-7}$ |
| 6 | 0.0625 | 0.025 | $-1.69 \times 10^{-7}$ | $-1.01 \times 10^{-7}$ |
| 7 | 0.0875 | 0.025 | $-1.65 \times 10^{-7}$ | $-9.85 \times 10^{-8}$ |
| 8 | 0.1125 | 0.025 | $-1.61 \times 10^{-7}$ | $-9.61 \times 10^{-8}$ |
| 9 | 0.1375 | 0.025 | $-1.57 \times 10^{-7}$ | $-9.37 \times 10^{-8}$ |
| 10 | 0.175 | 0.05 | $-1.51 \times 10^{-7}$ | $-9.03 \times 10^{-8}$ |
| 11 | 0.23 | 0.06 | $-1.43 \times 10^{-7}$ | $-8.54 \times 10^{-8}$ |
| 12 | 0.29 | 0.06 | $-1.35 \times 10^{-7}$ | $-8.04 \times 10^{-8}$ |
| 13 | 0.35 | 0.06 | $-1.27 \times 10^{-7}$ | $-7.58 \times 10^{-8}$ |

process have also been recently developed (e.g., Braud et al., 2009; Haverd and Cuntz, 2010). In this work the fractionation process was incorporated into the CATHY model using the sink term approach described above, setting 38 cm as the soil depth at which the maximum $^2H$-tracer concentration occurs. The injection term $f_c$ introduced into the transport equation is now modified such that there is no tracer mass re-injection in the first layer, and the amount re-injected progressively increases from

5    $qc/12$ to $qc$ between layers 2 and 13 (Table 2).

Besides the surface boundary, we set up a seepage face condition at the 23 x 30 nodes that constitute the downslope lateral boundary. For the transport equation the seepage face nodes have a zero Neumann (dispersive) assigned flux so that $^2H$ is allowed to exit the domain through advection with the outflowing water. All other LEO boundaries (the three other lateral boundaries and the base of the hillslope) were set to a zero Neumann condition for both the flow and transport equations (with

10   a zero water flux this implies that the advective flux for the transport equation is also zero).

The time stepping for the flow model is adaptive (based on convergence of the iterative scheme used to linearize Richards' equation (1)) and we set the time step range between $10^{-4}$ s and 90 s. The results in terms of velocity and saturation values





**Table 2.** Fractionation source term $f_c$ added to the transport equation and applied to the top 13 layers of the LEO grid. $f_{c1i}$ is the source term applied between rain pulses 1,2, and 3 while $f_{c2i}$ is the source term after the third pulse. $q$ is the sink term reported in Table 1.

| Layer | $f_c$ (1/s) | $f_{c1i}$ (1/s) | $f_{c2i}$ (1/s) |
|---|---|---|---|
| 1 | 0 | 0 | 0 |
| 2 | $-qc/12$ | $1.48{\times}10^{-8}\times c$ | $0.88{\times}10^{-8}\times c$ |
| 3 | $-qc/11$ | $1.59{\times}10^{-8}\times c$ | $0.95{\times}10^{-8}\times c$ |
| 4 | $-qc/10$ | $1.74{\times}10^{-8}\times c$ | $1.04{\times}10^{-8}\times c$ |
| 5 | $-qc/9$ | $1.90{\times}10^{-8}\times c$ | $1.14{\times}10^{-8}\times c$ |
| 6 | $-qc/8$ | $2.11{\times}10^{-8}\times c$ | $1.26{\times}10^{-8}\times c$ |
| 7 | $-qc/7$ | $2.30{\times}10^{-8}\times c$ | $1.40{\times}10^{-8}\times c$ |
| 8 | $-qc/6$ | $2.68{\times}10^{-8}\times c$ | $1.60{\times}10^{-8}\times c$ |
| 9 | $-qc/5$ | $3.14{\times}10^{-8}\times c$ | $1.87{\times}10^{-8}\times c$ |
| 10 | $-qc/4$ | $3.78{\times}10^{-8}\times c$ | $2.26{\times}10^{-8}\times c$ |
| 11 | $-qc/3$ | $4.70{\times}10^{-8}\times c$ | $2.85{\times}10^{-8}\times c$ |
| 12 | $-qc/2$ | $6.70{\times}10^{-8}\times c$ | $4.02{\times}10^{-8}\times c$ |
| 13 | $-qc$ | $1.27{\times}10^{-7}\times c$ | $7.58{\times}10^{-8}\times c$ |

were saved every 90 s or 900 s, respectively, during and between the rain events. These were linearly interpolated in time and read as input by the transport model, which was run with a fixed time step of 90 s for the entire simulation.

### 3.4 Simulations performed

The model simulations were used to interpret the integrated and point-scale flow and transport responses of the LEO hillslope. The guiding idea was to assess the need to increase the complexity of the model in progressing from first trying to reproduce the integrated flow response, then the integrated transport response, and finally the point-scale flow and transport responses. With the requirement that each new parameterization still had to satisfy the observation dataset from the previous level, the space of admissible solutions was progressively reduced. Initially the soil was assumed to be homogeneous and isotropic. The values of the van Genuchten parameters ($n_{vG}$=2.26, $\psi_{sat}$=-0.6 m, and $\theta_r$=0.002), the porosity ($n$=0.39), the saturated hydraulic conductivity ($k_s$=1.4$\times10^{-4}$ m/s), and the specific storage ($S_s$=5$\times10^{-4}$ m$^{-1}$) were obtained from laboratory analyses and simulations of prior LEO experiments (Niu et al., 2014; Pasetto et al., 2015). From this base set of parameter values for the first simulations, anisotropy, heterogeneity, and other variations were progressively introduced in the model.





**Table 3.** Configurations for the 6 simulations of the integrated flow analysis.

| Simulation | Saturated hydraulic conductivity (m/s) | | | Initial conditions | Rainfall |
|---|---|---|---|---|---|
| | Horizontal, $k_h$ | Vertical, $k_v$ | Seepage face, $k_{sf}$ | | |
| a | $1.4 \times 10^{-4}$ | $1.4 \times 10^{-4}$ | $1.4 \times 10^{-4}$ | Uniform | Spatially uniform |
| b | $6 \times 10^{-4}$ | $1.4 \times 10^{-4}$ | $1.4 \times 10^{-4}$ | Uniform | Spatially uniform |
| c | $6 \times 10^{-4}$ | $1.4 \times 10^{-4}$ | $2.2 \times 10^{-5}$ | Uniform | Spatially uniform |
| d | $6 \times 10^{-4}$ | $1.4 \times 10^{-4}$ | $2.2 \times 10^{-5}$ | Steady state | Spatially uniform |
| e | $6 \times 10^{-4}$ | $1.4 \times 10^{-4}$ | $2.2 \times 10^{-5}$ | Interpolated soil moisture measurements | Spatially uniform |
| e | $6 \times 10^{-4}$ | $1.4 \times 10^{-4}$ | $2.2 \times 10^{-5}$ | Interpolated soil moisture measurements | Spatially variable |

In the first step of this procedure (integrated flow response), we examined the influence of heterogeneity and anisotropy in saturated hydraulic conductivity (different $k_s$ at the seepage face and over the rest of the hillslope, on the basis of a clogging hypothesis from accumulation of fine particles (Niu et al., 2014); higher $k_h$ than $k_v$, on the basis of a hypothesis of slight vertical compaction leading to enhanced flow in the horizontal direction), rainfall (spatially uniform; spatially variable), and initial conditions (uniform; generated from a steady state simulation under drainage and evaporation; matching the soil moisture distribution at each sensor location). Six simulations were run in the first step. The configurations for each run are summarized in Table 3. For the initial conditions, in all three configurations (uniform for runs a through c, steady state for run d, and matching sensors for runs e and f), the same total initial water storage (26 m$^3$ as reported earlier) was used. For the atmospheric forcing, the spatially uniform rainfall rate (runs a through e) was the mean measured rate reported earlier (12 mm/h), while the spatially variable case (run f) was handled as described earlier. The evaporation rate, on the other hand, was kept spatially uniform for all 6 simulations and equal to the mean rate of 5.0 mm/d between the three pulses and 3.9 mm/d after the third pulse.

In the second step (integrated transport response), the effects of the dispersivity coefficients $\alpha_l$ and $\alpha_t$ and of isotope evaporation mechanisms on the amount of tracer at the seepage face outlet were explored. In the third step (flow point-scale data), the analysis focused on the soil moisture profiles obtained by averaging the observations and model results at specific depths (5, 20, 50, and 85 cm), and spatially variable (by layer) soil hydraulic properties ($n_{VG}$) were introduced. Finally, for the point-scale transport we compared the results obtained from some of the different parameterizations used in the previous steps.

The simulations performed are summarized in Table 4. Model performance was assessed against available observations using the coefficient of efficiency ($CE$) on seepage face flow $Q_{sf}$ for the integrated flow response and the root mean squared





**Table 4.** Simulation descriptions, parameter configurations, and performance metrics (coefficient of efficiency $CE$ and root mean squared error $RMSE$) for the integrated flow, integrated transport, and point-scale analysis steps.

| Integrated flow analysis | | | Simulations | | | | | $CE$ for $Q_{sf}$ |
|---|---|---|---|---|---|---|---|---|
| | | a | base case (Niu et al., 2014) | | | | | -0.62 |
| | | b | anisotropy | | | | | 0.64 |
| Effect on seepage | Effect on total | c | heterogeneity | | | | | 0.79 |
| face flow $Q_{sf}$ | water storage $V_s$ | d | initial conditions | | | | | 0.28 |
| | | e | initial conditions | | | | | 0.82 |
| | | f | rainfall distribution | | | | | 0.85 |
| **Integrated transport analysis** | | | $\alpha_l$ | evaporation | | | | $RMSE$ |
| | | g | 0.1 | all solute | | | | 0.12 |
| Effect on concentration $c$ at the seepage face | | h | 0.01 | all solute | | | | 0.037 |
| (flow configuration from simulation f) | | i | 0.001 | all solute | | | | 0.026 |
| | | j | 0.001 | fractionation | | | | 0.03 |
| | | k | 0.001 | no solute | | | | 0.045 |

| Point-scale analysis | | | | | | | | $RMSE$ for averaged $\theta$ (at 5, 20, 50, 85 cm depth) |
|---|---|---|---|---|---|---|---|---|
| Effect on averaged $\theta$ profiles | Effect on point-scale $\theta$ profiles | Effect on $c$ point-scale profiles (transport configuration from simulation i) | f | depth (cm) / $n_{VG}$ (homogeneous) | 5 | 20 / 2.26 | 50 | 85 | 10.36, 1.17, 1.73, 3.78 |
| | | | l | depth (cm) / $n_{VG}$ (heterogeneous) | 5 / 1.8 | 20 / 2.26 | 50 / 2.0 | 85 / 1.9 | 5.61, 1.43, 0.95, 1.72 |





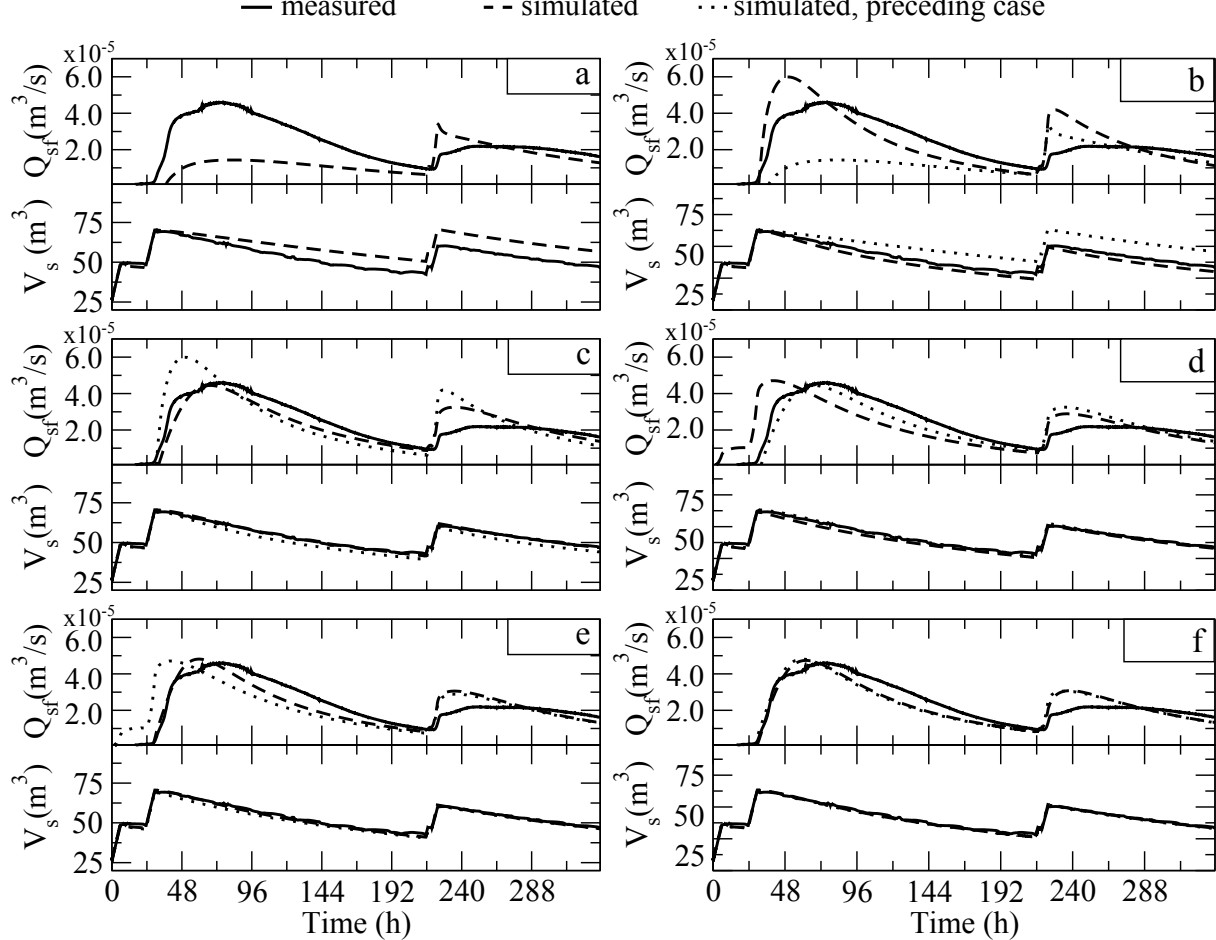

**Figure 3.** Results for the 6 simulations of the integrated flow response analysis (see Table 3). For each case the seepage face flow $Q_{sf}$ (top) and total water storage $V_s$ (bottom) are reported.

error ($RMSE$) on concentration $c$ at the seepage face for the integrated transport response and on averaged $\theta$ profiles for the flow point-scale response. The $CE$ and $RMSE$ metrics, also reported in Table 4, are calculated as in Dawson et al. (2007).

## 4 Results

### 4.1 Integrated flow response

5 In the first set of simulations we attempt to reproduce two integrated flow responses of the LEO hillslope, the measured seepage face flow and the measured total water storage. The results of the 6 simulations are presented in Fig. 3. The water balance partitioning between seepage face flow and internal storage was found to be strongly affected by the introduction of



anisotropy and heterogeneity in the hydraulic conductivity. We also found that the distribution of initial condition determines the timing of the first simulated seepage face peak and its shape. The spatial distribution of rain, on the other hand, was not found to have a significant impact on the model response. These general findings are described in more detail below.

In the first simulation (Fig. 3a), under the assumption of homogeneity, isotropy, uniform initial conditions, and spatially uniform rainfall and evaporation, the discrepancy between the simulated and observed response was large (a negative $CE$ is reported in Table 4), with the first and second peaks of the discharge hydrograph, respectively, underestimated and over-estimated by the model. In the second simulation, with the introduction of anisotropy (increasing the horizontal hydraulic conductivity $k_h$ to $6 \times 10^{-4}$ m/s), the overall model results for the seepage face flow improved notably compared to simulation a ($CE$ passed from -0.62 to 0.64) and the match for the total water storage was improved significantly (Fig. 3b). Next, the introduction of heterogeneity between the seepage face and the rest of the hillslope lowered the hydrograph peaks and smoothed out its overall shape (Fig. 3c), moving the simulated hydrograph closer to the measured one (and increasing $CE$ to 0.79). The effect of using distributed instead of uniform initial conditions is seen in comparing Figs. 3c, 3d, and 3e. Under uniform starting conditions the response was delayed in time, compared to the steady state case (generated under a drainage and evaporation run from initially wet conditions), where the response to the first rain pulse was faster. This faster response resulted in increased drainage due to longer recession periods, adversely affecting the match for the second pulse but improving the result for the third pulse. The simulation for Fig. 3e, with initial conditions closest to the initial state of the hillslope, resulted in a further increase in $CE$ to 0.82. For this run, the good match for the first hydrograph peak from simulation c of Table 3 was recovered, whilst retaining the good match for the second peak from simulation d. The simulated total water storage dynamics was already very well captured by simulation c and was not greatly affected by the initial conditions. The initial conditions from simulation e were used for all subsequent simulations discussed in this study. In the final simulation for the integrated flow response analysis, incorporating the spatial distribution of rainfall had a nominal impact on the results (Fig. 3f), with a slight increase in $CE$ to 0.85. Thus the actual distribution of atmospheric forcing, so long as it is not highly variable (which was part of the experimental design for the LEO tracer experiment), is less important than capturing the correct mean rate and total volume of these hydrologic drivers.

## 4.2 Integrated transport response

The velocity field and saturation obtained from the sixth flow simulation of the preceding section were used as input to the transport model. Figure 4 and Table 4 show, respectively, the results for the average tracer concentration at the seepage face and the $RMSE$ for different longitudinal dispersivity $\alpha_l$ values, namely 0.1 m, 0.01 m, and 0.001 m. The transverse dispersivity $\alpha_t$ was set one order of magnitude smaller than $\alpha_l$. The three graphs and the $RMSE$ values show that the discrepancy between the measured and simulated outflow concentration decreases with $\alpha_l$. At the highest value, $^2H$-labeled water appeared in the outflow discharge after the second pulse, whereas this did not occur in the measured data and in the model results for the model dispersivities. In all three cases the model reproduced the increase in tracer concentration after the last pulse, but whereas for $\alpha_l$=0.1 m the values were four times higher than the observed ones, for $\alpha_l$=0.01 m and $\alpha_l$=0.001 m they decreased significantly.





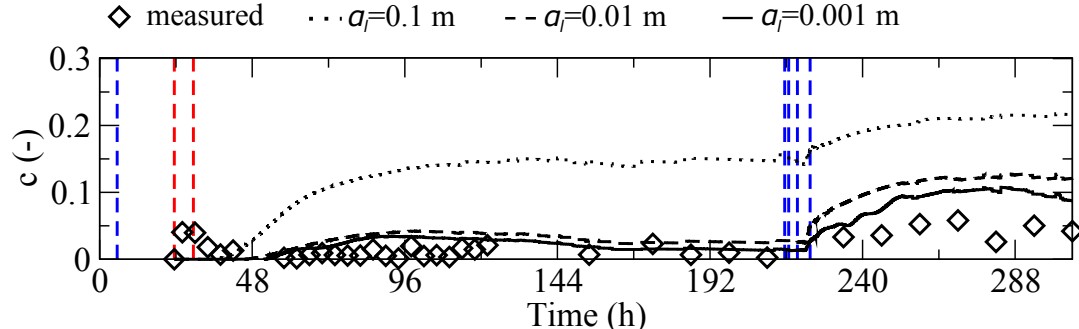

**Figure 4.** Results for the integrated transport response analysis for different values of dispersivity. The vertical dashed lines indicate the timing of the three pulses of rain (red when the water is $^2H$-enriched and blue when it is not).

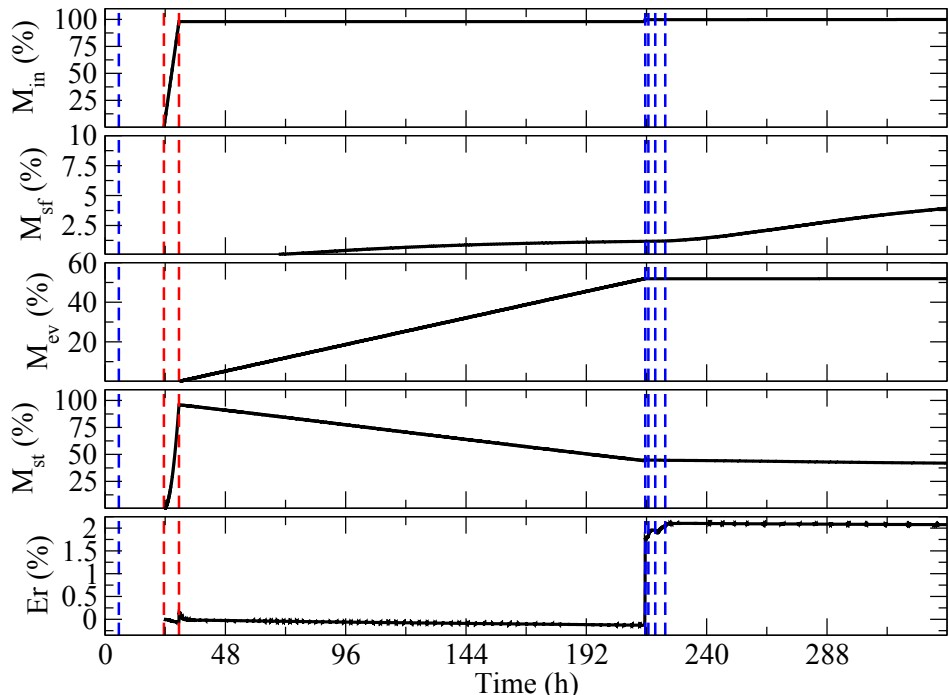

**Figure 5.** Simulated mass balance results for $\alpha_l$=0.001 m. From top to bottom: $^2H$ mass that enters the system, $M_{in}$ (normalized with respect to the total mass added to the system during the simulation); that exits through the seepage face, $M_{sf}$; that exits through evaporation, $M_{ev}$; and that remains in storage, $M_{st}$. The bottom graph shows the cumulative mass balance error $E_r$=($M_{in} - M_{sf} - M_{ev} - M_{st}$). The vertical dashed lines indicate the timing of the three pulses of rain (red when the water is $^2H$-enriched and blue when it is not).

The simulation using the lowest value of dispersivity was able to reproduce reasonably well the integrated measure of tracer response for the LEO hillslope.





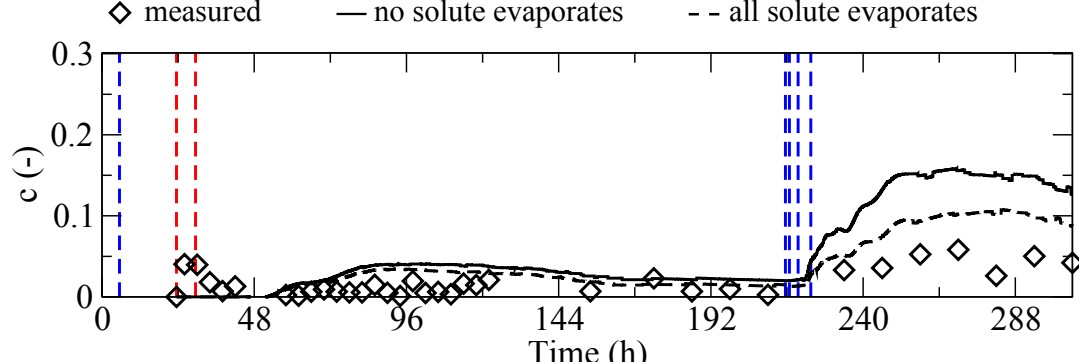

**Figure 6.** Measured and modeled average tracer concentration at the seepage face for the cases in which no solute and all solute leaves the system with evaporation. Both simulations are run for $\alpha_l$=0.001 m and $\alpha_t$=0.0001 m. The vertical dashed lines indicate the timing of the three pulses of rain (red when the water is $^2H$-enriched and blue when it is not).

To assess model accuracy, we report in Fig. 5 the mass balance results for the $\alpha_l$=0.001 m case, in terms of a balance between the cumulative mass of deuterium that entered the hillslope (with the second rainfall pulse), that exited the system (through seepage face outflow and evaporation), and that remained in storage. The results show that for $\alpha_l$=0.001 m and $\alpha_t$=0.0001 m, at the end of the simulation (after 14 d), 52% of the mass of $^2H$ injected has been lost through evaporation, about 4% has
seeped out, and the rest remained in storage, minus a cumulative mass balance error of about 2% with respect to the total mass injected. The high evaporative component computed by the model is a direct outcome of the zero dispersive flux surface boundary condition for the transport equation, through which any tracer in solution with evaporating water is advected away with the water. We examine next the impact of the sink term treatment of solute exchange across the land surface boundary, preventing any isotope tracer from evaporating.
The results of the sink term simulation in terms of average seepage face tracer concentration and mass balance are reported, respectively, in Fig. 6 and 7. As expected, the seepage face concentration has now increased, but only slightly, compared to the previous simulation. In mass terms, the seepage component has increased from 4% to 8% by the end of the simulation. With no tracer mass now exiting via the landscape surface, it is found instead that much more of the mass has remained in storage (about 90% compared to about 40% when the tracer was allowed to evaporate with the water). This result strongly suggests that
the tracer does not percolate far (deep) into the hillslope, perhaps as a result of the very dry conditions initially and during the whole experiment. A negative consequence of not allowing any tracer mass to evaporate, combined with low percolation, is an intense accumulation of the mass near the landscape surface, with tracer concentrations as high as 15. Further investigation is needed to understand whether this phenomenon is physically realistic or a numerical artifact. A compromise between allowing zero or all isotope tracer to leave the system via evaporation is to introduce isotopic fractionation processes into the model.
The results of the isotope fractionation simulation are reported in Figs. 8 and 9, respectively, for the average tracer concentration at the seepage face and the model mass balance results. The curve for the average concentration in Fig. 8 justly lies between the curves obtained by making all and no isotope evaporate with water. The mass balance shows that at the end of the





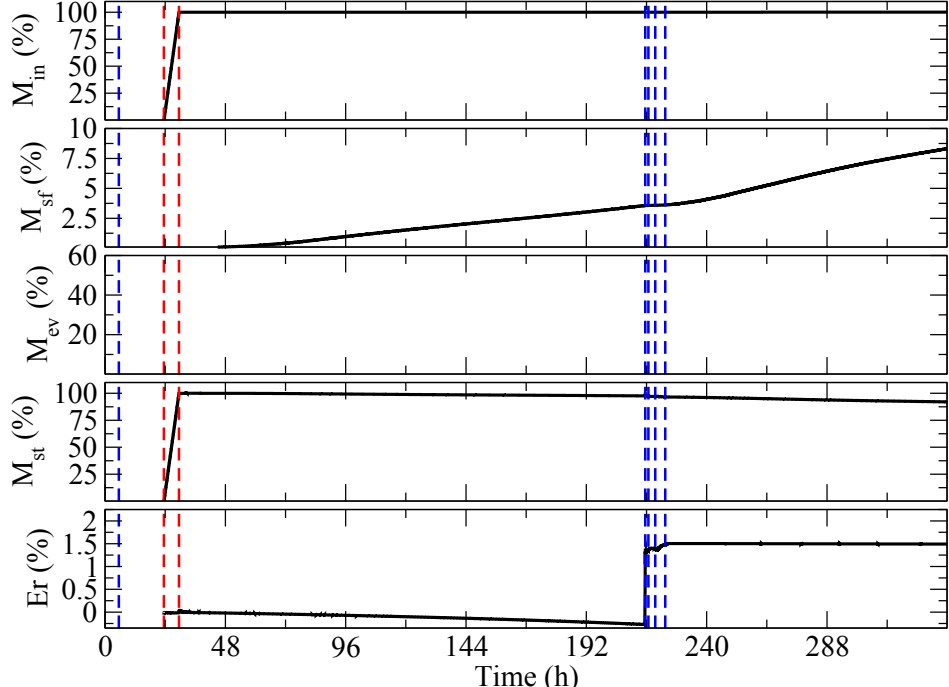

**Figure 7.** Simulated mass balance results for $\alpha_l$=0.001 m when the sink term is used to perform evaporation and the correction term $f_c$ added to the transport equation is used to force all the isotopic mass to stay in the system. From top to bottom: $^2H$ mass that enters the system, $M_{in}$ (normalized with respect to the total mass added to the system during the simulation); that exits through the seepage face, $M_{sf}$; that exits through evaporation, $M_{ev}$; and that remains in storage, $M_{st}$. The bottom graph shows the cumulative mass balance error $E_r$=($M_{in} - M_{sf} - M_{ev} - M_{st}$). The vertical dashed lines indicate the timing of the three pulses of rain (red when the water is $^2H$-enriched and blue when it is not).

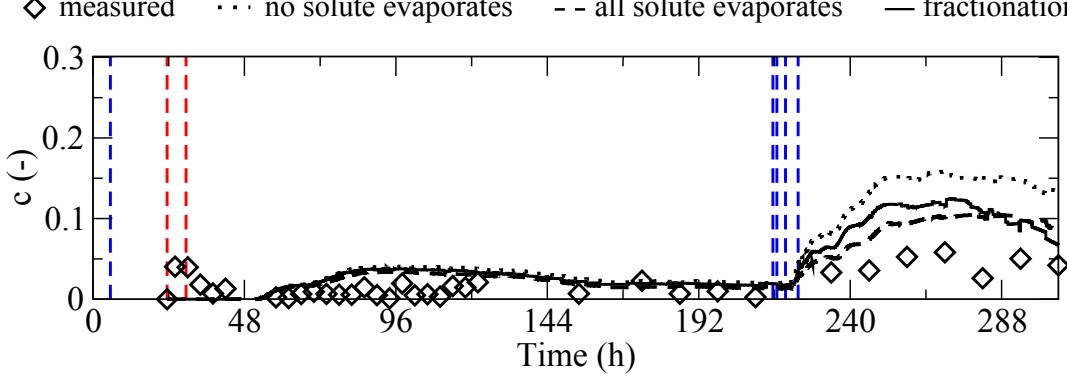

**Figure 8.** Measured and modeled average tracer concentration at the seepage face for the cases in which no solute, all solute, and some solute (fractionation) leaves the sytem with evaporation. The three simulations are run for $\alpha_l$=0.001 m and $\alpha_t$=0.0001 m. The vertical dashed lines indicate the timing of the three pulses of rain (red when the water is $^2H$-enriched and blue when it is not).





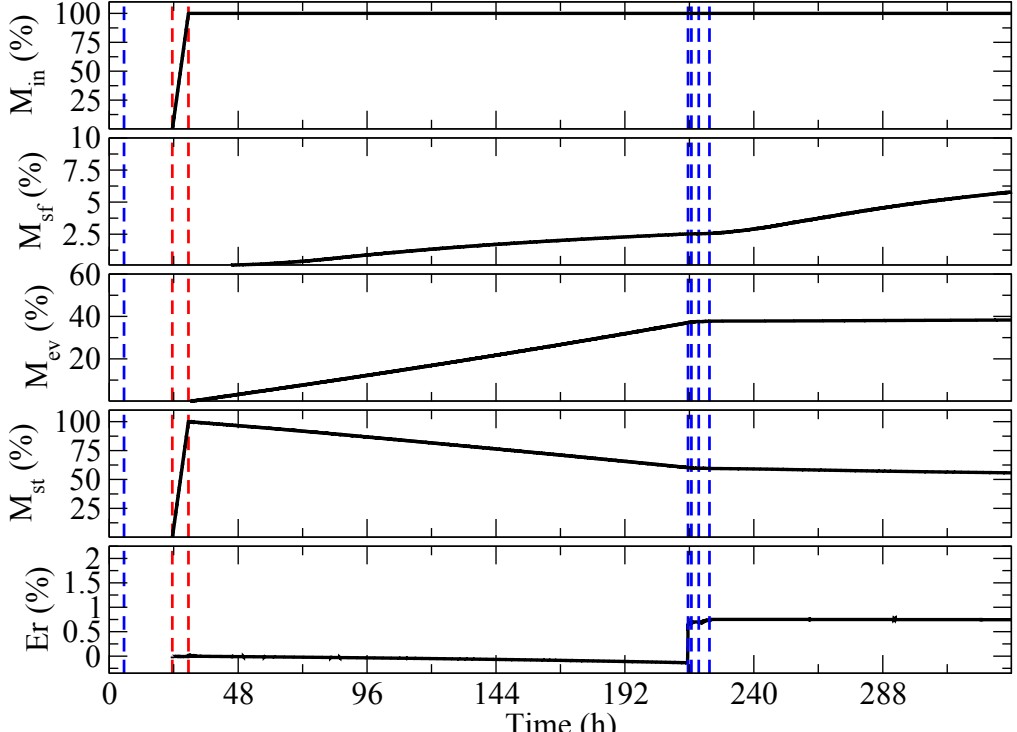

**Figure 9.** Simulated mass balance results for $\alpha_l$=0.001 m when the source term $f_c$ is added to the transport equation to perform isotopic fractionation. From top to bottom: $^2H$ mass that enters the system, $M_{in}$ (normalized with respect to the total mass added to the system during the simulation); that exits through the seepage face, $M_{sf}$; that exits through evaporation, $M_{ev}$; and that remains in storage, $M_{st}$. The bottom graph shows the cumulative mass balance error $E_r$=($M_{in} - M_{sf} - M_{ev} - M_{st}$). The vertical dashed lines indicate the timing of the three pulses of rain (red when the water is $^2H$-enriched and blue when it is not).

simulation 6.5% of the total mass injected has gone out through the seepage face, this result also falling between the previous simulations where zero or all isotope tracer was lost via evaporation. As expected, the evaporative mass loss is now significant (38%), but not as high as obtained when evaporation was treated as a land surface Neumann boundary condition (52%). The final mass balance error (0.75%) is lower than for the two previous simulations, and the accumulation of isotope mass just

5    below the land surface that occurred in the preceding case was not observed in this simulation.

## 4.3    Distributed flow response

For the distributed flow response analysis we first examined the behavior in time of the averaged soil water content value at the 4 depths of the sensor network (5, 20, 50, and 85 cm). That is, we compared the average of all sensor measurements at a given depth to the average of all simulated nodal $\theta$ values at that depth. The graphs for the results of simulation f from Table 3

10    (the configuration that best retrieved the integrated flow response) are shown in Fig. 10, while the $RMSE$ values are reported




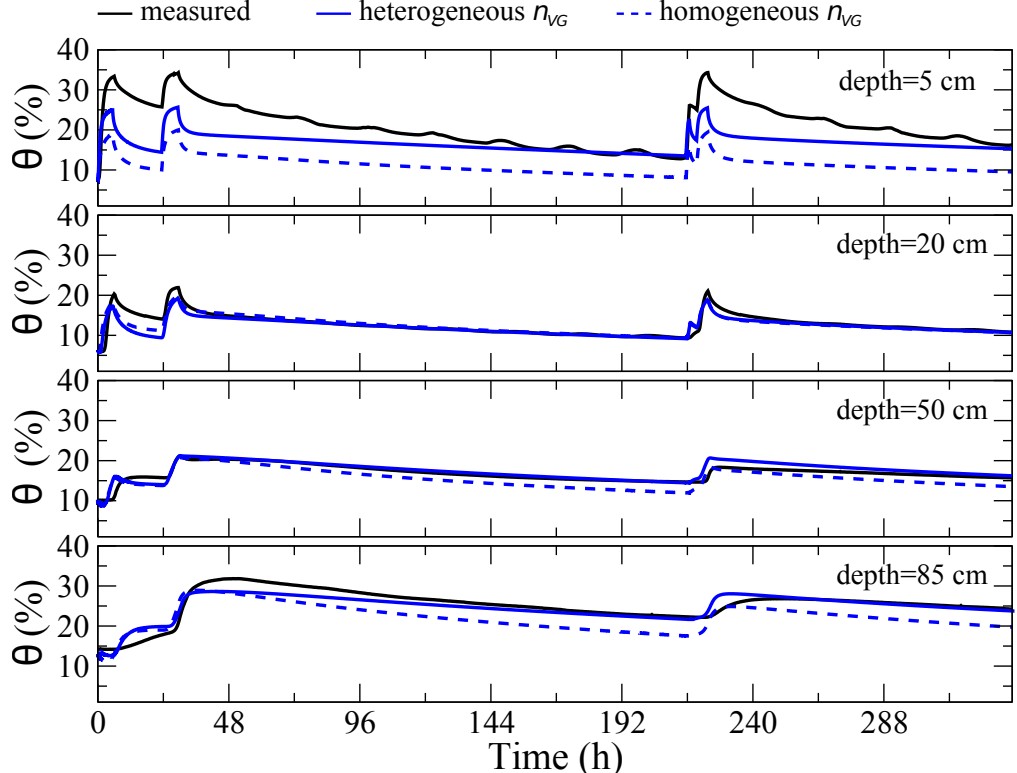

**Figure 10.** Averaged $\theta$ profiles at 5, 20, 50, and 85 cm depth from the surface.

in Table 4. The results show that at 50 cm there is a small underestimation by the model and that the model does not perform well at 5 cm and 85 cm compared to the profile at 20 cm.

To address this problem we increased the retention capacity of the soil by reducing, selectively, the $n_{VG}$ parameter of the van Genuchten hydraulic functions as reported in Table 4. We subdivided the soil profile into 4 strata encompassing the 4 sensor depths, and we decreased $n_{VG}$ for the strata closest to the surface (from 0 to 10 cm, $n_{VG}$=1.8), from 32 to 68 cm ($n_{VG}$=2.0), and from 68 cm to bottom ($n_{VG}$=1.9). For the second stratum (from 10 cm to 32 cm) the retention parameter was left unaltered from all previous simulations ($n_{VG}$=2.26) since the model already captured the observed response for the sensor at 20 cm depth quite well. The highest retention capacity (lowest $n_{VG}$ value) was set in the first stratum since the observation data show that the water content close to the landscape surface remains quite high, both during infiltration and drainage. The $n_{VG}$ values for the 4 strata reported here are the combination, from many trials, that best retrieved the observed averaged $\theta$ profiles. The results of this simulation are also shown in Fig. 10 and reported in Table 4. Compared to the results of the homogeneous $n_{VG}$ case, the model response improves significantly for the average profile at 5, 50, and 85 cm.

To take the distributed flow response analysis further, in Fig. 11 we show the water content time series at the four specific points shown in Fig. 2, at 5, 20, 50, and 85 cm depth from the surface. Sensor data at each of the 4 points and for each of the 4



**Figure 11.** Distributed (internal state) hydrological response for the $\theta$ profiles at 5, 20, 50, and 85 cm depth from the surface for four locations on the LEO-1 hillslope: point a (top left), point b (top right), point c (bottom left), and point d (bottom right) of Fig. 2.

soil depths is compared against both the homogeneous $n_{VG}$ case (simulation f from Table 3) and the heterogeneous $n_{VG}$ case (different value for each of the four strata). Once again the more detailed parameterization (variable $n_{VG}$) gives better results, although for some of the soil depths (in particular at 50 cm and 85 cm) and for some of the points (in particular point c) the discrepancies between simulated and measured $\theta$ time series are quite marked. It should be remarked that we did not run, as we did for the simulation summarized in Fig. 10, repeated trials to find a best fit, so it may perhaps be possible to optimize the fits against both the averaged $\theta$ data (Fig. 10) and the point data (Fig. 11) by manipulating the soil retention capacity for the 4 strata. However, it seems more likely that in going from a distributed but nonetheless averaged response variable to a distributed, point-scale response variable, additional model parameter complexity is needed to obtain an adequate response for all individual response variables.





**Figure 12.** Distributed (internal state) hydrological response for the tracer concentration breakthrough curves at 5, 20, 50, and 85 cm depth from the surface for four locations on the LEO-1 hillslope: point a (top left), point b (top right), point c (bottom left), and point d (bottom right) of Fig. 2. There were no tracer concentration measurements at 5 cm depth for point c and at 5 and 20 cm depth for point d. The transport model is run for $\alpha_l$=0.001 m and $\alpha_t$=0.0001 m. The vertical dashed lines indicate the timing of the three pulses of rain (red when the water is $^2H$-enriched and blue when it is not).

## 4.4 Distributed transport response

For the distributed transport response analysis we compared, as we did in Fig. 11 for the internal state flow response, the model results at individual points (a, b, c, d from Fig. 2) and individual soil depths (5, 20, 50, and 85 cm) for simulations using uniform (corresponding to configuration f from Table 3) and spatially variable (4 strata) soil retention capacity. The results are
5    shown in Fig. 12, and it can be seen that the model does not perform well at several locations within the hillslope (consistently at 20 cm depth, and at 5 cm depth for point b). Encouragingly, however, there is consistency with the previous distributed flow





results, in that the variable $n_{VG}$ run performs noticeably better than the spatially uniform case. For instance, with variable $n_{VG}$ the results are improved at the bottom of the hillslope, at 50 cm (for points b and c the modeled response gets closer to the measurements particularly after the third flush), and slightly at 5 cm (for point a).

For the distributed transport analysis we did not examine averaged concentration profiles at each of the 4 sensor depths (as we did for soil water content in Fig. 10) due to insufficient data. The sampling time and laboratory analysis costs for exhaustive measurement of isotopic compositions were prohibitive, thus there are much less data available for the distributed transport analysis compared to the flow case. The data gaps are also evident in Fig. 12: there are no measurements for 3 of the graphs, and scarce data at 50 cm depth for points a and d. It is also important to note that no additional parameterization was attempted for the distributed transport analysis. The main explicit parameters in the transport equation are the dispersivity coefficients, and these were experimented with in the integrated transport analysis. The transport equation is of course strongly dependent on flow velocities, and thus implicitly on the conductivity and other soil hydraulic parameters that were assessed in the flow model analyses. These and other parameterization issues will be further discussed in the next section.

To complete the sequence of analyses from integrated flow and then transport to distributed flow and transport, we used the simulation results from the additional parameterization introduced for the distributed analyses (spatially variable soil retention capacity) to assess model performance against the integrated flow and transport responses. The results (Fig. 13) show that while the match against tracer concentration at the seepage face has somewhat improved (compare with Fig. 8), the match against both of the integrated flow responses (seepage outflow and total water storage) has significantly deteriorated (compare with simulation f of Fig. 3). This is not a surprising result, given that no attempt was made to parameterize the model in tandem against both integrated and point-scale observations (nor against joint flow and transport data); the implications will be discussed below.

## 5 Discussion

Mass transport in unsaturated soils is extremely important in the context of biosphere, critical zone, and Earth systems research because of exchanges of water and solutes that occur across the land surface interface. However, its simulation is also known to be a particularly complex problem, compounded by any presence of heterogeneity. Wilson and Gelhar (1981), for instance, showed that spatial variations in moisture content affect solute plume spreading even without dispersive mixing, and that the rates of solute displacement are typically much smaller than the rates of moisture displacement. Birkholzer and Tsang (1997) demonstrated significant channeling effects (preferential solute pathways, with accompanying higher dispersion) at the extremes of very low saturation and full saturation. Studies that have combined comprehensive experimental observation with detailed subsurface simulation have also documented some of the challenges faced in modeling solute transport under unsaturated and heterogeneous conditions (e.g., Haggerty et al., 2004; Zheng et al., 2011). In this context, for the tritium and bromide tracer experiments at the Las Cruces trench site, standard models gave good prediction of wetting front movement during infiltration but poor prediction of point soil water content and tracer transport (Hills et al., 1991; Wierenga et al., 1991). For the macrodispersion (MADE) experiment, Russo and Fiori (2009) found that heterogeneity further enhances solute




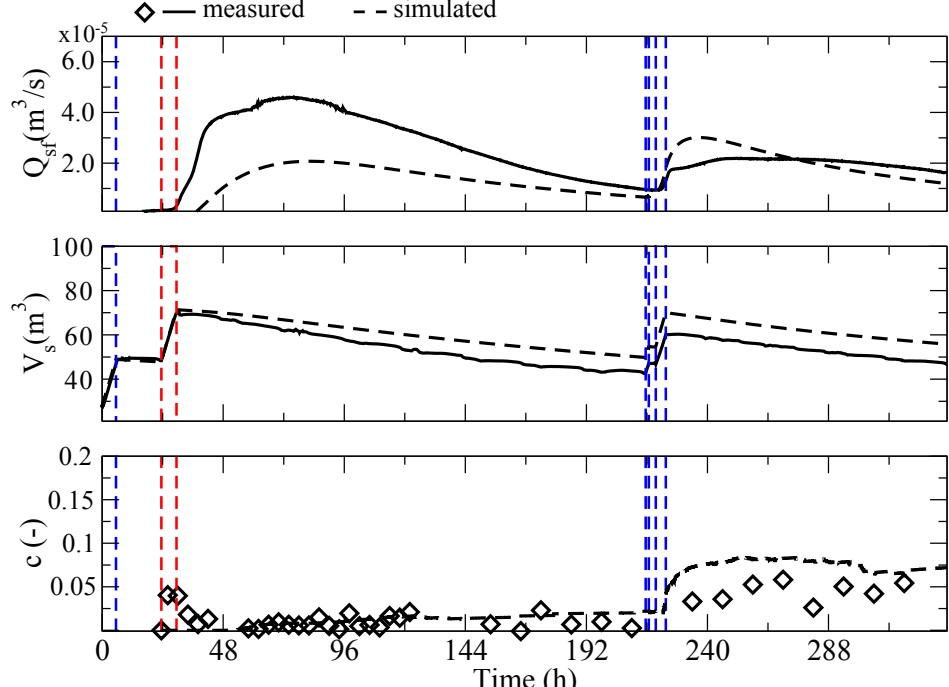

**Figure 13.** Performance of the model against integrated flow and transport responses (seepage face flow $Q_{sf}$, total water storage $V_s$, and average tracer concentration $c$ at the seepage face) using the additional parameterization from the distributed analyses (spatially variable soil retention capacity). The vertical dashed lines indicate the timing of the three pulses of rain (red when the water is $^2H$-enriched and blue when it is not).

spreading and breakthrough curve arrival times when the unsaturated zone is relatively dry or deep. In the present study, the additional heterogeneity introduced for the point-scale responses (namely spatially variable soil retention capacity) did not match as favorably the integrated (flow) observation dataset (Fig. 13). While this could perhaps be remedied using more rigorous or quantitative parameter estimation, the particular difficulties in capturing the point-scale concentration profiles,

5 especially near the landscape surface, can be taken as further evidence for flaws or gaps in theoretical understanding and model formulation (process representation) for simulating solute transport phenomena in very dry, heterogeneous soils.

  Various hypotheses have been invoked to explain possible factors that affect the migration and distribution of solutes under unsaturated, heterogeneous conditions, including: turbulent mixing due to high rainfall (Havis et al., 1992); solute transfer between mobile and immobile water (De Smedt and Wierenga, 1984); mobile-immobile exchange and hysteresis (Butters

10 et al., 1989; Russo et al., 1989a, b, 2014); lateral mixing due to velocity fluctuations (Russo et al., 1998); isotope effects (Barnes and Allison, 1988; LaBolle et al., 2008; Zhang et al., 2009); variable, state-dependent anisotropy (McCord et al., 1991); non-Gaussian early-time mean tracer plume behavior (Naff, 1990); non-Fickian solute migration at low water contents (Padilla et al., 1999) and for macroscopically homogeneous sand (Bromly and Hinz, 2004); and saturation-dependent dispersivity (Raoof and Hassanizadeh, 2013). In addition, Konikow et al. (1997) and Parker and van Genuchten (1984) discuss the importance





of boundary condition treatment (e.g., water-solute injection, solute exchange between soil and atmosphere). Given the many open questions, for this first analysis of the LEO isotope tracer experiment the modeling was kept to the standard formulation of the Richards and advection-dispersion equations. Limitations encountered in the multiresponse performance assessment, from the standpoint of experimental procedure, model formulation, or numerical implementation, will inform follow-up studies at

LEO. The simulation results from this tracer experiment, for instance, point to highly complex effects on plume migration of spatially variable water content in the dry soils that characterized the experiment, especially at early times.

## 6  Conclusions

In this study we have used multivariate observations (soil moisture, water and tracer outflow, breakthrough curves, and total water storage) culled from the first isotope tracer experiment at the LEO-1 hillslope of the Biosphere 2 facility to explore

some of the challenges in modeling unsaturated flow and transport phenomena. Integrated (first flow and then transport) and distributed (again flow followed by transport) measurements were progressively introduced as response variables with which to assess the results from simulations with CATHY, a 3D numerical model for variably saturated flow and advective-dispersive solute migration. Compared to the first flow experiment at LEO that was successfully modeled with CATHY (Niu et al., 2014), the modeling task for the tracer experiment was significantly more complicated due to: joint simulation of both flow and

transport processes; considerably drier initial conditions and reduced forcing; performance assessment against both system-wide and point-scale measurements; and multiple periods of water/tracer injection compared to a single rainfall episode. In some sense the previous flow study looked at the first order response of the LEO hillslope, whereas the modeling exercise for the tracer experiment represents a first look at higher order responses of the Biosphere 2 landscapes.

There are several findings from this first set of simulations of a LEO isotope tracer experiment. At the start of the exercise,

where integrated flow measurements were used, we were able to obtain good matches for two response variables (total water storage and seepage face outflow) using parameter values and initial and boundary conditions that correspond quite closely to the actual experimental conditions and previous (flow experiment) model implementation (Niu et al., 2014). The same soil parameterization was successfully used to reproduce the integrated transport response. When passing to point-scale flow and finally point-scale transport, a refinement of the model setup (augmenting the degree of heterogeneity, mainly) was needed.

Moreover, providing more information to the model (for example, the distribution of initial water storage rather than just the initial total volume) generally helped to improve the simulation results.

The effect of saturated hydraulic conductivity (heterogeneity and anisotropy) on the response of subsurface hydrologic models is well known, and was also borne out in this study. Also not surprisingly, the dispersivity parameter had a big impact on the transport simulations, with a clear trend to a better match against measured seepage face concentration as dispersivity

was decreased. The spatial distribution of rainfall was not found to have a big impact on simulation results, and there was not much difference, in terms of isotope tracer mass exiting the seepage face, between the zero, partial, and no fractionation cases, suggesting that the injected tracer did not percolate very far into the hillslope, likely due to the very dry initial conditions.



We conclude with a few specific recommendations for alleviating some of the modeling and experimental limitations encountered during this study. On the modeling side, a more sophisticated treatment of solute transport phenomena beyond the standard advection-dispersion equation could start with incorporation of a mobile-immobile conceptualization and/or saturation-dependent dispersivity. Other upgrades to the CATHY model (e.g., Scudeler et al., 2016) will mitigate the grid Peclet constraint

and provide more reliable flow velocity calculations, essential to maintaining low mass balance errors and high accuracy in solute transport. On the experimental side, higher solute concentrations (including labeled tracers), wetter initial conditions, and more intensive direct or indirect measures of total solute mass could help address the high sensitivity of solute transport to small scale heterogeneity under dry soil conditions. Any experiments that provide spatially detailed observations of both flow and transport response variables that are then jointly used in estimating, for instance, conductivity and other soil hydraulic

parameters (traditionally identified based solely on flow responses), would be critical to advancing the present state of hydrologic model verification, given the high impact that Darcy velocities, which are directly dependent on such parameters, have on solute mixing processes. Finally, future LEO isotope tracer experiments that also generate some surface runoff would offer valuable benchmark data for improving integrated surface-subsurface models.

*Acknowledgements.* This work was supported in part by the National Science Foundation (NSF grants EAR-1344552 and EAR-1417097).

The authors wish to thank Nate Abramson, Anouk Gevaert, Ed Hunt, Michael Pohlmann, Michael Sibayan, and Yvonne Smit for technical support. The Biosphere 2 facility and the capital required to conceive and construct LEO were provided through a charitable donation from the Philecology Foundation, and its founder, Mr. Edward Bass. We gratefully acknowledge this charitable donation. The data for this paper are available by contacting the corresponding author.



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
