# Peer review of "Multiresponse modeling of variably saturated flow and isotope tracer transport for a hillslope experiment at the Landscape Evolution Observatory"

_Hydrology and Earth System Sciences, 2016_

## Referee Comment (RC1) · C. Mugler (Referee) · 17 Jun 2016

REVIEW OF "MULTI-RESPONSE MODELING OF AN UNSATURATED ZONE ISOTOPE TRACER EXPERIMENT AT THE LANDSCAPE EVOLUTION OBSERVATORY" (HESS-2016-228)

SUMMARY:

This manuscript deals with the modeling of an unsaturated flow and isotope tracer experiment. The experiment, conducted at the Landscape Evolution Observatory (LEO), involved successive injections of water and deuterium-enriched water into an initially very dry hillslope. Multivariate observations were presented for flow and transport: soil moisture, water and tracer outflow, breakthrough curves and total water storage. Simulations were performed with the physically-based distributed numerical model CATHY that solves the 3D Richards and advection-dispersion equations and includes coupling with surface routing equations. The modeling approach succeeded in simulating the integrated flow and transport responses. However, with the same parameterization it failed to restitute the point measurements of the water contents and the tracer concentrations.

OVERALL QUALITY:

This manuscript is clear, well structured, and pleasant to read. The experimental results are new. However, they should be better described and discussed. It is surprising to see that these well-calibrated experiments are so difficult to model. Some of the numerous parameterizations added in the successive simulations look arbitrary and their choice should be better justified. Furthermore, the cumulative mass balance error of tracer in the CATHY simulations is relatively large (~2% with respect to the total mass injected) and this fact should therefore be discussed. The conclusions of the manuscript would be more convincing if more than one numerical code were used. But this task could be further accomplished in a future publication. Very surely, these experiments and their first simulations could serve as a nice benchmark for physically-based distributed numerical models provided the full dataset is rendered available.

In my opinion, the experimental results and the corresponding simulations are very interesting and deserve to be published in HESS. However, some corrections and/or clarifications should be accomplished prior to publication. The authors will find below some remarks to correct or complete their manuscript.

MAJOR COMMENTS:

(1) The experimental results are new and interesting. However, the description and discussion of the water contents and concentrations measured should be improved. You will find below some examples of questions that arise about the experimental results.

(1a) Page 4, Figure 1: Please comment the peak of $\delta^2 H$ during the first irrigation event.

(1b) Page 18, line 7: You are using the soil water content at 4 different depths averaged over 496 sensors. Can you quantify the soil heterogeneity from a statistical analysis of these 496 experimental vertical profiles?

The landscape geometry is symmetric. All parameter heterogeneities included in the simulations are symmetrical as well. Does one observe this symmetry also in the experimental results? For example, are the $\theta$ vertical profiles measured along two vertical lines that are located at the same distance from the seepage face but on either side of the landscape similar?

Is the variability of the profiles correlated with the rainfall variability?

(1c) Page 20, Figure 11: The time evolutions of $\theta$ measured at the points located at the centre of the hillslope (points a, b, and c) clearly show that the bottom of the hillslope has become water-saturated after the 2[nd] rainfall event. This point should be discussed in the paper. CATHY has clearly failed to simulate this saturated zone. More generally, practically all $\theta$ values obtained from the simulation, presented in Fig.11 are lower than the corresponding values obtained from the measurements. Do you have an explanation for this lack of water in the CATHY simulations? I think it would be better to calibrate some parameters (e.g., $n_{VG}$) from distributed $\theta$ profiles instead of calibrating parameters from averaged $\theta$ profiles. As a matter of fact, the vertical evolution of the wetting front varies depending on whether it is observed at the top of the landscape or in the zone of flow convergence.

(2) Some hypotheses and some results of the modeling approach require further argumentation and discussion.

(2a) Page 8, line 16: Several parameterizations in the simulations are arbitrary and not justified. For example, why did you choose a depth of 38 cm? Did you perform a calibration? Evaporation is often assumed to be active only over the first few centimeters.

(2b) Page 8, line 21: Same as remark (2a): Why did you choose $\lambda=1$ m$^{-1}$?

(2c) Page 8, line 20: There is no moisture content dependence term in the sink term given by Eq. (15). What happens if there is not enough water for evaporation in the upper 38 cm of soil?

(2d) Page 9, lines 3-5: Same as remarks (2a) and (2b): the choice of $f_c$ looks very arbitrary. Please justify it.

(2e) Page 11, Table 3: Same as remarks (2a), (2b), and (2d): how did you choose the $k$ values? Did you perform a calibration?

(2f) Page 11, lines 1-6: The heterogeneity and anisotropy of $k_s$ are justified by invoking the processes of clogging and compaction. Such modifications should induce a modification of the topography. Did you observe topographic changes caused by diffusive geomorphic processes such as rain splash during the rain events that lasted several hours? Pangle et al. (2015) affirm that "digital elevation models will be constructed at regular intervals and following all events with the potential to modify the topography", with a model surface precision of 0.002 m. Have you performed such measurements? If yes, did you observe some changes in the topography? Have you observed the formation of some crusts at the soil surface? The properties of the soil, e.g., its permeability, must be changed with time if crusts are forming.

(2g) Page 14, line 28: The best numerical results are obtained with the smallest value of the dispersivity. Can you discuss this result? Is it a proof that the soil is very homogeneous? It would be interesting to measure $\alpha_L$ for example from transport experiments in a column filled with the same porous media.

(2h) Page 16, line 5: Can you explain why the cumulative mass balance error is so large (~2%)?

**MINOR COMMENTS**:

(3) Page 4, line 3: Please clearly indicate the location of the seepage face. Is it the 11-m$^2$ boundary at the downslope end of the landscape?

(4) Page 4, line 9: The estimated evaporation rates are two times and ten times larger than the rates reported in Niu et al. (2014) and Pasetto et al. (2015), respectively, although the soil is drier. Can you explain this difference?

(5) Page 4, Figure 1: The irrigation rate is equal to 12 mm/h ~ $1.1 \times 10^{-3}$ $m^3$/s. Please correct the y-scale for $Q_r$ in Fig. 1.

(6) Page 4, Figure 1: Does the size of the symbols for $\delta^2H(t)$ correspond to the 0.5‰ analytical precision?

(7) Page 5, Equation (1): The CATHY model solves the coupling between surface and subsurface flows. Why do you not quote the surface flow equation?

(8) Page 5-6, section 3.2: How are the nonlinear terms in the equations being solved? Is it based on an iterative scheme with Picard iterations?

(9) Page 6, line 17: Please remove Eq.(6) because the effective saturation has already been defined in line 20.

(10) Page 6, line 19: Please replace the exponent in Eq.(8) with "-m" and add the definition of m:
$m = 1 - 1/n_{vG}$.

(11) Page 6-7, Eqs. (9)-(13): I am not convinced of the interest to present Equations (9)-(13). What is new in comparison with the schemes already described by Putti et al. (1998) or by Weill et al. (2011)?

(12) Page 7, section 3.3: How did you choose the horizontal and vertical discretizations? Did you verify the spatial convergence of the numerical simulations?

(13) Page 8, line 23: Please correct the values for the evaporation:
5 mm/d ≡ $5.8 \times 10^{-8}$ m/s and 3.9 mm/ d ≡ $4.5 \times 10^{-8}$ m/s

(14) Page 9, Caption of Table 1: $z_i$ is the depth of the middle of the $i^{th}$ layer.

(15) Page 9, Table 2: Please verify the values given in Table 2.
For example, for layer 5, $f_{c1i}=1.91\times10^{-8}\times c$, for layer 7, $f_{c1i}=2.36\times10^{-8}\times c$ and $f_{c2i}=1.41\times10^{-8}\times c$, for layer 11, $f_{c1i}=4.77\times10^{-8}\times c$, for layer 12, $f_{c1i}=6.75\times10^{-8}\times c$.

(16) Page 11, Table 3: Please correct the name of the last simulation: "f" instead of "e".

(17) Page 11, line 19: Please provide the definition of the coefficient of efficiency CE.

(18) Page 13, Figure 3: Figure 3 would be clearer if the time evolutions of the seepage face flow and of the total water storage for a given case were reported side by side instead of one above the other. Furthermore, the superposition of two simulated test cases in each figure is unnecessary.

(19) Page 14, lines 19-20 and line 26: Finally, which simulation (e or f) is used for the subsequent simulations? Please correct the text accordingly.

(20) Page 14, lines 30-32: I do not understand what you mean. In my opinion, in Fig.4, $^2$H-labeled water appears in the measured outflow discharge and also in all simulated outflow discharges after the second pulse.

(21) Page 16, line 11: You cannot claim that a ~50% increase of the seepage face concentration after the third event is a slight increase.

(22) Page 16, line 17: The definition of c implies: 0<c<1. What do you mean by a tracer concentration as high as 15? It would be interesting to show some vertical profiles of the water content and concentration.

(23) Page 18, line 2: In the first simulation, a part of the isotope tracer may evaporate but it is not all lost by evaporation.

(24) Page 20, line 2, and Page 21, line 4: Please add the name of the simulation: simulation l from Table 4. More generally, indicate in all figures the name of the simulations as specified in Table 4.

---

## Referee Comment (RC2) · Anonymous Referee #2 · 4 Jul 2016

A review of: Multiresponse modeling of an unsaturated zone isotope tracer experiment at the Landscape Evolution Observatory, by Scudeler et al.

Overview

The manuscript describes in detail efforts to fit a variably-saturated flow and convection-dispersion transport models to a very highly controlled field-scale experiment in a 1:1 physical analog of a hillslope. The physical models and numerical approximations are described in mathematical terminology (e.g. zero Neumann for no flow boundary condition etc.) yet readable for a wide community of hydrologists. The beauty of the paper is in the clear description of the need to increase the complexity of the model in the process of fitting first the integrated flow response (in which unique

transient observations of seepage face flow-rate, and total storage of water are available in this experimental system) than the integrated transport response, than further complexity is needed to fit point observations of water content and concentrations. The model does not go very far with complexity, it starts in uniform hydraulic properties, moving to different properties near the seepage face and layered porous medium but does not go further to variability within layers, or mobile immobile formulations etc.. Hence, the well-known, good fit of the macro phenomenon relatively to poorer fits to point observations is described very clearly.

Recommendation

I am not sure there is completely new modeling knowledge here, nevertheless, the paper has "educational quality" for hydrological modelers as well as very unique experimental data (although not in focus in the manuscript), and therefore, I warmly recommend publication in HESS, following the authors pay their attention to the comments herein.

Major Comments

1) Title – The hillslope problem as well as the model used here and the results of the experiment, are variably saturated rather than unsaturated (saturation at 85 cm for significant duration of the experiment in most locations, Figure 11). Suggest to change to: Multiresponse modeling of variably saturated flow and isotope tracer transport in a hillslope experiment at the Landscape Evolution Observatory.

2) Discussion - In line with the previous comment. I don't understand why the authors do not discuss the more specific setup of a hillslope that was studied here, rather than concentrating on general unsaturated flow. The hillslope case has significant differences than the general unsaturated zone (variably saturated, lateral flow component dominant, relations with evaporation and runoff etc.). Many simulation studies of hillslopes can be discussed (e.g. Fiori and Russo, 2008 WRR).

3) List of symbols – There are many symbols in equations and within the text. For example, it took me too long to find what does the nee in line 27 page 7 stands for. I suggest adding a list of symbols at the beginning of the paper.

4) Use of the term heterogeneity – is misleading. Changing a homogenous model deterministically to have lower Ks near the seepage face, or different hydraulic properties at different layers doesn't make it a heterogeneous model (a term now used for a medium in which the properties vary from pixel to pixel randomly usually constrained to a PDF and a spatial correlation function). I suggest describing this type of additional complexity with different (more explicit) terms (e.g. low Ks at seepage face, layered n(vg) etc.), throughout the text, tables and figures.

5) Fractionation? - in water isotopes during evaporation. The term fractionation is brought up late in the methods section (page 8) as if it is totally trivial. I suggest to add a paragraph on fractionation of water isotopes during evaporation in the introduction to introduce the topic before jumping into the details of dealing with modeling it in the methods section.

6) Van Genuchten (1985) - should be van Genuchten (1980). It would have been a specific comment for any other paper in hydrology (p. 6, l. 15 and in reference list).

Specific Comments

1) P. 4, Figure 1.  a) Lowest pane (delta2H) – zoom into the interval of interests in the vertical axis (< 53); b) say something on the high readings at the beginning before tracer introduction, and just before the third rain pulse.  Or looking at Figure 4 there seems to be a shift of the data to the left? Solve the problem, explain.

2) P. 5, l. 14, Eq. 2. I suggest to add the sink\source term – f(c) to the 2H transport equation here as well, rather than only elaborating on it in table 2 and related text.

3) P. 7, l. 27. Shouldn't the left hand side of the equation be n (or theta)*v*nee, rather than only v*nee (porous medium approximation of ratio of flux and velocity).

4) P. 10 l. 5-8. Excellent lines – don't touch, makes it so much easier to follow the long descriptions after.

5) P. 11, l. 11. The evaporation rates – were they calculated from the water balance and the load cell data? Or how? Please elaborate.

6) P. 15 Figure 4. A) Say something on the early breakthrough during the heavy isotope injection. B) Elaborate in the text why was the high disprsivity simulation so much biased upwards in the mass of tracer exiting the system (earlier arrival times are expected in high dispersity but also late ones. What were the left-in-storage or evaporated components of the mass balance in the high dispersity run?

7) P. 16, Figure 6 and related text: solute is not a proper term for 2H.

Please also note the supplement to this comment:
http://www.hydrol-earth-syst-sci-discuss.net/hess-2016-228/hess-2016-228-RC2-supplement.pdf

---

## Referee Comment (RC3) · Anonymous Referee #3 · 8 Jul 2016

**Summary:**

This manuscript provides detailed comparisons of multiple hydrologic response variables using a sophisticated integrated hydrology model and highly controlled experiment at the Landscape Evolution Observatory. The authors experiment with different levels of complexity within the model and demonstrate the importance of model heterogeneity if the goal of the model is to match spatially distributed points as opposed to integrated responses. Results also indicate the importance of considering more than just integrated hydrologic response variables when determining model parameters.

**Recommendation:**

Overall I find the paper to be well written. I think it provides an interesting comparison of a state of the art experiment with state of the art modeling that will be interesting to the hydrologic community and should be published in HESS. I find their scientific approach to be sound; however, I do think that some changes to the manuscript to better outline all of the test cases and highlight differences would make the discussion easier to follow. I also think that the manuscript would be of broader interest if the authors would devote some discussion the relevance of these findings to other commonly used or similar modeling approaches. I have provided detailed suggestions to this effect below.

**Major Comments:**

1. The introduction is focused on the need for multi objective parameter optimization. This is a good motivator for this work, but also the study is not really presenting advances for parameter optimization. Rather it's evaluating the impact of different parameterizations on model response. Therefore, I think it would be helpful to provide more background on heterogeneity and variably saturated flow processes and the state of the practices for both modeling and observations. I think this would provide a better context for where both the modeling and observations used here compare to previous work.

2. I would appreciate more details on why the observational experiments were setup the way they were. For example, how were the rainfall rates and timing determined?

3. It can be hard to keep all of the different simulations setups straight throughout the paper. I think this could be addressed by expanding on Figure 2 to better label different aspects of the domain that are discussed in the model setup and creating a new table or conceptual model that summarizes all of the runs in one place.

4. The discussion of differences between basins is mostly qualitative. I think some additional figures that plot differences between scenarios for key metrics and discussion points would strengthen the conclusions.

5. This study uses the CATHY model, but it is focuses on addressing larger questions in model uncertainty and parameterizations. Given this goal I think some additional discussion on the degree to which these results are specific to the model you are using or would be universal to other integrated flow and transport models would be quite helpful.

**Specific Comments:**

1. Page 2, line 8: Please expand on this point. What do you mean by 'an important example of this complexity'? Are you saying that parameter estimation has been particularly challenging for mass transport?
2. Page 3, line 6: Clarify, "infrastructure" for what?
3. Page 3, line 10: From this description it sounds like a simple sloping slab but from Figure 2 it appears that it is actually a tilted v sloping to the center of the domain. Please clarify. Also you could annotate the slopes on Figure 2 to make this even more clear.
4. Page 4, line 2: You should clarify that you are talking about just the rain from the first event here not 'all the rain water'
5. Page 4, line 2: Also here you switch from using the term 'irrigation' to 'rain'. It will be easier to follow if you pick one term and stay consistent.
6. Page 7, line 14: Please expand here to clarify how you decided on this lateral resolution.
7. Page 7, line 25: This is a very dense and long sentence. In my opinion it would easier to follow and refer back to if this information were provided in the form of a table. Also, if you keep this in paragraph form you should tie the three numbered experiments listed to simulations a-f in Table 3?
8. Page 8 line 16: How did you determine the 38cm depth for evaporation? This seems arbitrary.
9. Page 9 line 6: It would be helpful to have visual on your model figure for where the seepage face is occurring.
10. In my opinion the source sink terms listed in Tables 1 and 2 would be more easily interpreted graphically. Alternatively, I'm not sure that this information is necessary for the interpretation of the results as long as you describe how you got these terms so potentially these tables could also be deleted.
11. Table 3: Why is simulation e repeated twice in this table
12. Table 4 is difficult to follow. I think you need a separate table describing the setup of runs g-l and then report only the output metrics in this table. Also, it might help to just focus on runs g-l here and add the information for simulations a-f to Table 3.
13. Figure 3: Please describe what 'simulated, preceding case' means in the caption.
14. Figures 4, 6 and 8: I think the diamonds for the measured values should be smaller so that they are not overlapping each other or the axes so much.

---

## Author Comment (AC1) · 15 Jul 2016

**Response to Reviewer comments for manuscript HESS-2016-228**, "Multiresponse modeling of an unsaturated zone isotope tracer experiment at the Landscape Evolution Observatory", by Carlotta Scudeler, Luke Pangle, Damiano Pasetto, Guo-Yue Niu, Till Volkmann, Claudio Paniconi, Mario Putti, and Peter A. Troch

**SUMMARY:**

This manuscript deals with the modeling of an unsaturated flow and isotope tracer experiment. The experiment, conducted at the Landscape Evolution Observatory (LEO), involved successive injections of water and deuterium-enriched water into an initially very dry hillslope. Multivariate observations were presented for flow and transport: soil moisture, water and tracer outflow, breakthrough curves and total water storage. Simulations were performed with the physically-based distributed numerical model CATHY that solves the 3D Richards and advection-dispersion equations and includes coupling with surface routing equations. The modeling approach succeeded in simulating the integrated flow and transport responses. However, with the same parameterization it failed to restitute the point measurements of the water contents and the tracer concentrations.

**OVERALL QUALITY:**

This manuscript is clear, well structured, and pleasant to read. The experimental results are new. However, they should be better described and discussed. It is surprising to see that these well calibrated experiments are so difficult to model. Some of the numerous parameterizations added in the successive simulations look arbitrary and their choice should be better justified. Furthermore, the cumulative mass balance error of tracer in the CATHY simulations is relatively large (~2% with respect to the total mass injected) and this fact should therefore be discussed. The conclusions of the manuscript would be more convincing if more than one numerical code were used. But this task could be further accomplished in a future publication. Very surely, these experiments and their first simulations could serve as a nice benchmark for physically-based distributed numerical models provided the full dataset is rendered available. In my opinion, the experimental results and the corresponding simulations are very interesting and deserve to be published in HESS. However, some corrections and/or clarifications should be accomplished prior to publication. The authors will find below some remarks to correct or complete their manuscript.

1) We wish to thank the Reviewer for the attention to our work and the very detailed and constructive comments. The main issues raised above are taken up individually below and we will respond to each point raised. We agree that the data from the LEO experiments would make nice modeling benchmarks; the dataset from this paper is indeed available, as noted in the acknowledgements.

**MAJOR COMMENTS:**

(1) The experimental results are new and interesting. However, the description and discussion of the water contents and concentrations measured should be improved. You will find below some examples of questions that arise about the experimental results.

(1a) Page 4, Figure 1: Please comment the peak of  $d^2H$  during the first irrigation event.

2) Please note that we have made a mistake in that plot. The curve should be shifted by 23.5 h (see Figures 4, 6, 8, and 13). Thus the peak appears with the second pulse of rain, in the early seepage face flow. This is probably due to the fact that the residual soil water in the landscape prior to irrigation had become somewhat enriched in deuterium (compared to the irrigation water) during evaporation. In fact, during evaporation, hydrogen will preferentially go into the vapor phase

compared to deuterium, so that the liquid phase remaining in the soil easily becomes slightly enriched in deuterium. The delta d2H values in that early seepage flow may reflect some mixing of the new irrigation water with the evaporatively-enriched residual soil moisture. This slight enrichment may disappear in the seepage flow at later times just due to dilution of that residual soil moisture by the newly infiltrating irrigation water. We are going to add this information to the section of the paper that describes the experiment.

**(1b) Page 18, line 7: You are using the soil water content at 4 different depths averaged over 496 sensors. Can you quantify the soil heterogeneity from a statistical analysis of these 496 experimental vertical profiles?**

3) The detailed information generated from the LEO experiments constitutes a valuable dataset for analyses such as the one suggested by the reviewer. It should be possible, albeit not within the scope of the present work, to perform inverse modeling to retrieve conductivity distributions based on this moisture information (note: in reality the 496 sensors do not correspond to 496 vertical profiles, since at each sampling position there are 2 to 4 sensors, at different depths). In addition to soil heterogeneity, there are other factors that can affect the soil moisture response at the different locations (heights and positions) of the hillslope. In the two figures below we plot the standard deviation for both the observed and modeled profiles. For the modeled response we have done this for both the homogeneous  $n_{vg}$  and heterogeneous  $n_{vg}$  cases. From this analysis we can see that the deviations from the average profiles for the observed and modeled responses are similar (apart from the results at 85 cm depth), suggesting that the model parameterization is quite adequate. We propose to replace Figure 10 in the original manuscript with these two more detailed figures, and to revise the description of these results accordingly.

Figure 1. Averaged soil water content ( $\theta$ ) profiles at 5, 20, 50, and 85 cm (top to bottom) depth from the surface: observed (solid black curves) and calculated (solid blue curves, simulation f). In each graph the deviation from the mean (one standard deviation above and below) is shown as dashed lines (blue for the model and black for the measurements).

---

## Author Comment (AC2) · 18 Jul 2016

**Response to Reviewer comments for manuscript HESS-2016-228**, "Multiresponse modeling of an unsaturated zone isotope tracer experiment at the Landscape Evolution Observatory", by Carlotta Scudeler, Luke Pangle, Damiano Pasetto, Guo-Yue Niu, Till Volkmann, Claudio Paniconi, Mario Putti, and Peter A. Troch

*Overview*

*The manuscript describes in detail efforts to fit a variably-saturated flow and convection-dispersion transport models to a very highly controlled field-scale experiment in a 1:1 physical analog of a hillslope. The physical models and numerical approximations are described in mathematical terminology (e.g. zero Neumann for no flow boundary condition etc.) yet readable for a wide community of hydrologists. The beauty of the paper is in the clear description of the need to increase the complexity of the model in the process of fitting first the integrated flow response (in which unique transient observations of seepage face flow-rate, and total storage of water are available in this experimental system) than the integrated transport response, than further complexity is needed to fit point observations of water content and concentrations. The model does not go very far with complexity, it starts in uniform hydraulic properties, moving to different properties near the seepage face and layered porous medium but does not go further to variability within layers, or mobile immobile formulations etc.. Hence, the well-known, good fit of the macro phenomenon relatively to poorer fits to point observations is described very clearly.*

*Recommendation*

*I am not sure there is completely new modeling knowledge here, nevertheless, the paper has "educational quality" for hydrological modelers as well as very unique experimental data (although not in focus in the manuscript), and therefore, I warmly recommend publication in HESS, following the authors pay their attention to the comments herein.*

We wish to thank the Reviewer for the attention to our work. The major and specific comments raised by the Reviewer are addressed below.

*Major Comments*

*1) Title – The hillslope problem as well as the model used here and the results of the experiment, are variably saturated rather than unsaturated (saturation at 85 cm for significant duration of the experiment in most locations, Figure 11). Suggest to change to: Multiresponse modeling of variably saturated flow and isotope tracer transport in a hillslope experiment at the Landscape Evolution Observatory*

1) We like the title proposed by the Reviewer and will adopt it, changing only "in a hillslope" to "for a hillslope".

*2) Discussion - In line with the previous comment. I don't understand why the authors do not discuss the more specific setup of a hillslope that was studied here, rather than concentrating on general unsaturated flow. The hillslope case has significant differences than the general unsaturated zone (variably saturated, lateral flow component dominant, relations with evaporation and runoff etc.). Many simulation studies of hillslopes can be discussed (e.g., Fiori and Russo, 2008 WRR).*

2) In the Discussion section we wished to draw attention to the numerous challenges in modeling solute transport (rather than flow) phenomena in the unsaturated zone, given that this is the area that in our opinion raised the most difficulties – physical and numerical – in our modeling of the LEO experiment, in particular with regards to capturing point-scale responses. We did not make a specific

distinction between field, hillslope, and (small) catchment-scale studies, as we feel that the issues and proposed explanatory hypotheses (last paragraph of the Discussion) apply across the board. Nonetheless, taking the cue from the Reviewer's concern about a hillslope focus and the mention of the Fiori and Russo paper, we will add a mention of transit time distribution research at the hillslope scale, as this is an excellent example of the need for better modeling of flow and transport dynamics and a very active area of current research. The first two sentences of the Discussion section will thus become three and will read: "Mass transport in unsaturated soils is extremely important in the context of biosphere, critical zone, and Earth systems research because of exchanges of water and solutes that occur across the land surface interface. The study of hillslope transit time distributions (e.g., Fiori and Russo, 2008; Botter et al., 2010; Heidbüchel et al., 2013; Tetzlaff et al., 2014) is a good example of the need for a better understanding of such water and solute exchanges and the consequent subsurface flowpaths. The simulation of unsaturated zone mass transport phenomena is however known to be a particularly complex problem, …".

*3) List of symbols – There are many symbols in equations and within the text. For example, it took me too long to find what does the nee in line 27 page 7 stands for. I suggest adding a list of symbols at the beginning of the paper.*

3) We do not think a list of symbols is warranted for a standard-length paper. We will however examine all the places in the paper where symbols are used, and replace or supplement these symbols with the variable names to serve as reminders and thus to make the paper more readable.

*4) Use of the term heterogeneity – is misleading. Changing a homogenous model deterministically to have lower Ks near the seepage face, or different hydraulic properties at different layers doesn't make it a heterogeneous model (a term now used for a medium in which the properties vary from pixel to pixel randomly usually constrained to a PDF and a spatial correlation function). I suggest describing this type of additional complexity with different (more explicit) terms (e.g. low Ks at seepage face, layered n(vg) etc.), throughout the text, tables and figures.*

4) We agree with this comment and will make the suggested changes throughout the paper (including the figures and tables).

*5) Fractionation? - in water isotopes during evaporation. The term fractionation is brought up late in the methods section (page 8) as if it is totally trivial. I suggest to add a paragraph on fractionation of water isotopes during evaporation in the introduction to introduce the topic before jumping into the details of dealing with modeling it in the methods section.*

5) Although we totally agree that fractionation is not a trivial topic, it is nonetheless not a main topic of the paper. We therefore prefer not to emphasize it too strongly in the Introduction, as this would entail having to also describe the other configurations that were tested (this is all done in Section 3.3). We thus propose, rather than adding an entire paragraph on fractionation, to add the following sentence at the very end of the Introduction: "The boundary condition configurations, for instance, includes a sink-based treatment of isotope fractionation to allow only a portion of the tracer to evaporate with the water."

Although we totally agree that fractionation is not a trivial topic, it is nonetheless not a main topic of the paper. We therefore prefer to introduce it as is currently done, in Section 3.3 as one of the surface

boundary condition options that we investigate.

*6) Van Genuchten (1985) - should be van Genuchten (1980). It would have been a specific comment for any other paper in hydrology (p. 6, l. 15 and in reference list).*

6) Thanks for catching this. The error maybe crept in because we often cite the van Genuchten and Nielsen 1985 paper for these constitutive relationships. We will correct "1985" to "1980".

*Specific Comments*

*1) P. 4, Figure 1. a) Lowest pane (delta2H) – zoom into the interval of interests in the vertical axis (< 53); b) say something on the high readings at the beginning before tracer introduction, and just before the third rain pulse. Or looking at Figure 4 there seems to be a shift of the data to the left? Solve the problem, explain.*

1) Indeed there is an error in Figure 1, thanks for making us realize this. The delta$^2$H graph should be shifted by 23.5 h, as is evident from Figures 4, 6, 8, and 13. The corrected figure is shown below. The high readings straight after the beginning of the tracer introduction is a point that was also raised by the Reviewer 1. The peak in the early seepage face flow is probably due to the fact that the residual soil water in the landscape prior to irrigation had become somewhat enriched in deuterium (compared to the irrigation water) during evaporation and reflects some mixing of the new irrigation water with the evaporatively-enriched residual soil moisture. We will add this information to the section of the paper that describes the experiment. Note that in the corrected graph there is no longer a high reading before the third rain pulse. As suggested we have also rescaled the y-axis in this graph.

[Figure]

*2) P. 5, l. 14, Eq. 2. I suggest to add the sink\source term – f(c) to the 2H transport equation here as well, rather than only elaborating on it in table 2 and related text.*

2) Thanks for the suggestion. We will add two terms in the transport equation: $qc^*$ [M/TL$^3$], with $c^*=c$ if $q$ (the source/sink term in the flow equation 1) is a sink term, otherwise an imposed source concentration, and $f_c$, which is a generic source/sink term (our correction term used to model fractionation).

*3) P. 7, l. 27. Shouldn't the left hand side of the equation be n (or theta)\*v\*nee, rather than only v\*nee (porous medium approximation of ratio of flux and velocity).*

3) The boundary term arising from the finite element P1 Galerkin discretization of equation 1 (at the left hand side of the = sign) is:

$$ -\int_{\Gamma_f} K_r(\psi)K_s(\nabla\psi + \eta z) \cdot v \, \mathrm{d}\Gamma_f $$

where $\Gamma_f$ is the Neumann boundary of the domain for the flow. Thus, the equation in line 27 is correct. You can find more details on the flow equation discretization in Scudeler et al. (2016), cited in the paper.

*4) P. 10 l. 5-8. Excellent lines – don't touch, makes it so much easier to follow the long descriptions after.*

4) Thanks.

*5) P. 11, l. 11. The evaporation rates – were they calculated from the water balance and the load cell data? Or how? Please elaborate.*

5) The evaporation rates were calculated from the seepage face measurement and the load cell data. In particular, being $V_{sf}$ the cumulative volume flowing out from the seepage face between two events or after the last event until the end of the experiment (information directly obtained from the flow meter measurements), being $dV$ the change in water volume between two events or after the last event until the end of the experiment (information directly obtained from the load cell data), being $dT$ the time interval between two events or after the third event until the end of the experiment, the average evaporation rate was calculated as $(-V_{sf}+dV)/dT$. We will add this explanation on page 4 after line 10.

*6) P. 15 Figure 4. A) Say something on the early breakthrough during the heavy isotope injection. B) Elaborate in the text why was the high dispersivity simulation so much biased upwards in the mass of tracer exiting the system (earlier arrival times are expected in high dispersivity but also late ones. What were the left-in-storage or evaporated components of the mass balance in the high dispersivity run?*

6) A) Do you mean for the measured response? If so we have answered this point above (see response to specific comment 1) and we are going to add that information in the revised manuscript. It is also true that the model response presents an early breakthrough after the injection of isotope as we have nonzero solute concentration at the seepage face also after the second pulse but the values are not as high as after the third pulse. These early breakthroughs may be due to dispersion effects. B) We show below the mass balance results for the simulation relative to the 0.1 m longitudinal dispersivity case. At the end of the simulation almost 40% of the mass injected has flown out from the seepage face, 16% has evaporated, and 42% has remained in storage (compared to, respectively,

~4%, 52%, and 42% of the $\alpha_l$=0.001 m case). 0.1 m is 1/10 of the depth of the hillslope, compared to 0.001 m and 0.01 which are only the 1/1000 and 1/100, respectively. The effect of the high dispersivity makes the solute percolate down quickly to then flow out of the domain from the seepage face boundary. As a consequence it is also less exposed to evaporation. We would expect another breakthrough with a fourth pulse of rain. Unfortunately, no measurements were taken after 336 h. We will not add the figure below to the paper but we will discuss a little bit more in detail the results for the 0.1 m case.

[Figure]

Simulated mass balance results for the $\alpha_l$=0.1 m case.

*7) P. 16, Figure 6 and related text: solute is not a proper term for $^2H$.*

7) We agree, hydrogen isotopes are not solutes. In the revised manuscript we will no longer refer to deuterium as a solute.

---

## Author Comment (AC4) · 25 Jul 2016

**Response to Reviewer comments for manuscript HESS-2016-228**, "Multiresponse modeling of an unsaturated zone isotope tracer experiment at the Landscape Evolution Observatory", by Carlotta Scudeler, Luke Pangle, Damiano Pasetto, Guo-Yue Niu, Till Volkmann, Claudio Paniconi, Mario Putti, and Peter A. Troch

*Summary:*

*This manuscript provides detailed comparisons of multiple hydrologic response variables using a sophisticated integrated hydrology model and highly controlled experiment at the Landscape Evolution Observatory. The authors experiment with different levels of complexity within the model and demonstrate the importance of model heterogeneity if the goal of the model is to match spatially distributed points as opposed to integrated responses. Results also indicate the importance of considering more than just integrated hydrologic response variables when determining model parameters.*

*Recommendation:*

*Overall I find the paper to be well written. I think it provides an interesting comparison of a state of the art experiment with state of the art modeling that will be interesting to the hydrologic community and should be published in HESS. I find their scientific approach to be sound; however, I do think that some changes to the manuscript to better outline all of the test cases and highlight differences would make the discussion easier to follow. I also think that the manuscript would be of broader interest if the authors would devote some discussion the relevance of these findings to other commonly used or similar modeling approaches. I have provided detailed suggestions to this effect below.*

We wish to thank the Reviewer for the attention to our work. The comments raised by the Reviewer are addressed below.

*Major Comments:*

*1. The introduction is focused on the need for multi objective parameter optimization. This is a good motivator for this work, but also the study is not really presenting advances for parameter optimization. Rather it's evaluating the impact of different parameterizations on model response. Therefore, I think it would be helpful to provide more background on heterogeneity and variably saturated flow processes and the state of the practices for both modeling and observations. I think this would provide a better context for where both the modeling and observations used here compare to previous work.*

1) We agree with the reviewer that the paper is not really about parameter optimization. In the same sense, neither is it really about heterogeneity, as we do not conduct a systematic analysis based on complex configurations and innumerable realizations. But parameter estimation and heterogeneity are certainly underlying themes of the paper, and we agree that a mention of heterogeneity in the context of variably saturated flow processes is warranted. In the Introduction we will add a sentence on this and include additional citations. Lines 7-9 of page 2 will become: "… to more complex models. Traditional challenges, on both experimental and modeling sides, are associated with soil heterogeneity, variability in parameters, and variably saturated conditions (e.g., Binley et al., 1989; Woolhiser et al., 1996; Neuweiler and Cirpka, 2005; see Reference below). An added source of

complexity arises when passing from flow modeling to flow and transport modeling (e.g., Ghanbarian-Alavijeh et al., 2012; Russo et al., 2014)." See also reply to specific comment 1 below.

*2. I would appreciate more details on why the observational experiments were setup the way they were. For example, how were the rainfall rates and timing determined?*

2) We will add in section 2 just after the sentence ending "…precipitation at rates between 2 and 40 mm/h." the following additional description of the rain system: "Each landscape at LEO has 5 independent plumbing circuits, each including a different array of sprinkler heads, and therefore generating a different irrigation flux." And in section 3.1 we will add after the first paragraph the following new paragraph providing additional details on the rainfall rates and timing: "At the time of this experiment we consistently used one plumbing circuit because the spatial distribution of irrigation produced by this circuit had been well characterized by in situ testing. This allowed us to examine the possible influence of spatially heterogeneous irrigation patterns on flow and transport. The purpose of the first irrigation application was to increase the average moisture content of the landscape, which had received no irrigation for more than 40 days prior. The second irrigation application was used to introduce the deuterium tracer. No additional irrigation was applied for multiple days so that the tracer transport within, and out of the landscape, would be affected by soil-moisture redistribution and evaporation. The third and final irrigation application was applied with the intention of forcing additional tracer mass beyond the seepage face boundary, to reveal additional detail in the measured breakthrough curve. In retrospect, and following laboratory analysis that spanned several weeks, we only observed the initiation of the tracer breakthrough curve at the seepage face."

*3. It can be hard to keep all of the different simulations setups straight throughout the paper. I think this could be addressed by expanding on Figure 2 to better label different aspects of the domain that are discussed in the model setup and creating a new table or conceptual model that summarizes all of the runs in one place.*

3) In Figure 2 we will indicate the seepage face and the atmospheric forcing boundary.
The revised figure is shown below. The only thing that differs in the model setup amongst the different simulations is the treatment of the atmospheric boundary. With the introduction of a new table containing the boundary condition setup for each simulation it should be easier to follow the setup of the different simulations (see response to specific comment 7 below).

[Figure]

Figure 2. 3D numerical grid for the LEO landscape. Points a, b, c, and d are the locations where samples were extracted during the experiment for subsequent laboratory analysis.

*4. The discussion of differences between basins is mostly qualitative. I think some additional figures that plot differences between scenarios for key metrics and discussion points would strengthen the conclusions.*

4) We are not sure we understand what is meant by 'differences between basins'. Does the reviewer mean the 3 different LEO hillslopes? The experiment being analyzed in this paper was conducted on just one of the 3 hillslopes. This is mentioned in the first sentence of section 3.1 ("…performed at the LEO-1 hillslope …") but we agree that this is not at all very clear. We will add the following clarification after the first sentence of the last paragraph of the Introduction: "Both of these experiments were performed on the first of the three hillslopes at LEO to be commissioned, hereafter referred to as LEO-1." We believe that the revisions to the main text and figures that we are bringing to the paper, in response also to the other two reviewers, will help strengthen the conclusions (see, for example, response 3 to Reviewer 1 and response 6 to the specific comments of Reviewer 2).

*5. This study uses the CATHY model, but it is focuses on addressing larger questions in model uncertainty and parameterizations. Given this goal I think some additional discussion on the degree to which these results are specific to the model you are using or would be universal to other integrated flow and transport models would be quite helpful.*

5) We agree with this remark and will add the following new paragraph at the end of the Discussion section: "The broad results of our study should be quite universal, particularly to deterministic numerical models based on the 3D Richards and advection-dispersion equations. However, any model has its specific features and differs, for example, in the way equations are coded (e.g., choice of numerical solvers) or interface conditions are implemented (e.g., free-surface vs boundary condition switching). For insights on the impact of specific model differences in the performance of CATHY-like models, see the intercomparison studies of Sulis et al. (2010; see References below) and Maxwell et al. (2014; already cited in the paper). These intercomparison studies have thus far focused only on flow processes, and there is an urgent need to extend the analyses to solute transport phenomena, in order to properly guide our assessment of the physical and numerical correctness of competing models as these models continue to increase in complexity. For instance for this study there are aspects of the CATHY model related to how we implemented evaporation and fractionation that might be expected to negatively impact the generality of our findings, although in terms of isotope tracer mass exiting the seepage face the impact was quite small. But the implementation here was somewhat ad hoc, and more study is needed on the importance and proper representation of fractionation in solute transport models, especially under strongly unsaturated conditions."

*Specific Comments:*

*1. Page 2, line 8: Please expand on this point. What do you mean by 'an important example of this complexity'? Are you saying that parameter estimation has been particularly challenging for mass transport?*

1) We are alluding here to the added complexity (more equations, more parameters, etc) when passing from flow to flow and transport modeling. We will clarify this sentence to: "An added source of complexity arises when passing from flow modeling to flow and transport modeling (e.g., Ghanbarian-Alavijeh et al., 2012; Russo et al., 2014)." This sentence is also made clearer with the new sentence added just before this one (see reply to major comment 1 above).

*2. Page 3, line 6: Clarify, "infrastructure" for what?*

2) We will change this to "research infrastructure".

*3. Page 3, line 10: From this description it sounds like a simple sloping slab but from Figure 2 it appears that it is actually a tilted v sloping to the center of the domain. Please clarify. Also you could annotate the slopes on Figure 2 to make this even more clear.*

3) In fact LEO consists of three v-shaped hillslopes. The average slope of each landscape is $10^o$, as stated in the paper, while the local slope varies from upslope positions to the convergence zones, with maximum slope of $17^o$ near the convergence zone. Since it is difficult to incorporate this information graphically in Figure 2, we will add this information to the text, in section 2.

*4. Page 4, line 2: You should clarify that you are talking about just the rain from the first event here not 'all the rain water'.*

4) We are talking about all rain water. The confusion is perhaps due to the phrase "… and generated seepage face outflow that started after 5 h" at the end of this sentence. We propose to split this sentence in two: "All the rain water applied infiltrated into the soil. Seepage face flow started 5 h after the beginning of the experiment."

*5. Page 4, line 2: Also here you switch from using the term 'irrigation' to 'rain'. It will be easier to follow if you pick one term and stay consistent.*

5) We agree that this is inconsistent and may cause confusion. Since we use the terms "rainfall" / "rain" / "precipitation" more than "irrigation" in the paper, we will use these former terms exclusively.

*6. Page 7, line 14: Please expand here to clarify how you decided on this lateral resolution.*

6) In the revised manuscript we will explain why we have chosen this discretization (see our response 23 to Reviewer 1, comment 12).

*7. Page 7, line 25: This is a very dense and long sentence. In my opinion it would easier to follow and refer back to if this information were provided in the form of a table. Also, if you keep this in paragraph form you should tie the three numbered experiments listed to simulations a-f in Table 3?*

7) We will add the suggested table (it will become Table 1 in the revised manuscript), shown below, and we will revise (shorten and simplify) the paragraph in question.

| Simulation (see Tables 4 and 5) | Rain with $^2$H-enriched water (second pulse) | | Rain with no $^2$H-enriched water (first and third pulses) | | Evaporation (between rain pulses and after the third pulse) | |
|---|---|---|---|---|---|---|
| a-f, g-i | Flow | Transport | Flow | Transport | Flow | Transport |
| | $q_n^f$=-12 mm/h | $q_c^t = v \cdot vc^*$, $c^*$=1 | $q_n^f$=-12 mm/h | $q_c^t$=0 | $q_n^f$=5 or 3.9 mm/h | $q_n^t$=0 |
| j | $q_n^f$=-12 mm/h | $q_c^t = v \cdot vc^*$, $c^*$=1 | $q_n^f$=-12 mm/h | $q_c^t$=0 | Sink $q$ (Table 1) | Source $f_c$ (Table 2) |
| k | $q_n^f$=-12 mm/h | $q_c^t = v \cdot vc^*$, $c^*$=1 | $q_n^f$=-12 mm/h | $q_c^t$=0 | Sink $q$ (Table 1) | Source $f_c$ with $f_c$=-$qc$ |

Table 1. Treatment of boundary conditions at the land surface during the rainfall and evaporation periods for the flow and transport models.

*8. Page 8 line 16: How did you determine the 38cm depth for evaporation? This seems arbitrary.*

8) This is a point that has also been raised by another reviewer (see our response 6 to comment 2a of Reviewer 1). The parameterizations were chosen in order to qualitatively reproduce the experimental results obtained by Barnes and Allison [1988], where it is shown that, for isotope profiles in unsaturated soil and under evaporation, the maximum concentration can also occur at 50 cm from the surface. Above this point the isotope concentration decreases rapidly towards the surface due to the diffusion of water vapor to the soil surface. In our model we assume that the region dominated by water vapor diffusion is also the one characterized by evaporation, and selected 38 cm for the threshold. We will describe this better in the revised manuscript.

*9. Page 9 line 6: It would be helpful to have visual on your model figure for where the seepage face is occurring.*

9) In the revised manuscript we will show graphically, in Figure 2, where the seepage face is set. The revised figure and caption are shown above (see response to major comment 3 above).

*10. In my opinion the source sink terms listed in Tables 1 and 2 would be more easily interpreted graphically. Alternatively, I'm not sure that this information is necessary for the interpretation of the results as long as you describe how you got these terms so potentially these tables could also be deleted.*

10) We agree and will replace Tables 1 and 2 with the figure shown below.

[Figure]

Figure. Sink term *q* and source term *fc* over depth *z* added to the flow and transport equation, respectively. *q1* and *fc1* are applied between rain pulses 1, 2, and 3, while *q2* and *fc2* are applied after rain pulse 3.

*11. Table 3: Why is simulation e repeated twice in this table*

11) This is a mistake, the last one should be "f".

*12. Table 4 is difficult to follow. I think you need a separate table describing the setup of runs g-l and then*

*report only the output metrics in this table. Also, it might help to just focus on runs g-l here and add the information for simulations a-f to Table 3.*

12) We think it is important to keep Table 4 since it summarizes all simulations performed and makes it easier to follow the text in both the Simulations performed subsection and in the Results section.

*13. Figure 3: Please describe what 'simulated, preceding case' means in the caption.*

13) The "preceding case" results will be removed from this figure (see our response 29 to Reviewer 1, comment 18, that also includes the new figure and caption).

*14. Figures 4, 6 and 8: I think the diamonds for the measured values should be smaller so that they are not overlapping each other or the axes so much.*

14) We agree. We show below the new Figure 4 with smaller diamonds for the measured values. We will do this also for Figures 1, 6, 8, 12, and 13.

[Figure]

**References**

Binley, A., J. Elgy, and K. Beven (1989), A physically based model of heterogeneous hillslopes: 1. Runoff production, Water Resour. Res., 25(6), 1219–1226, doi:10.1029/WR025i006p01219

Woolhiser, D. A., R. E. Smith, and J.-V. Giraldez (1996), Effects of spatial variability of saturated hydraulic conductivity on Hortonian overland flow, Water Resour. Res., 32(3), 671–678, doi:10.1029/95WR03108.

Neuweiler, I., and O. A. Cirpka (2005), Homogenization of Richards equation in permeability fields with different connectivities, Water Resour. Res., 41, W02009, doi:10.1029/2004WR003329.

Sulis, M., S. B. Meyerhoff, C. Paniconi, R. M. Maxwell, M. Putti, and S. J. Kollet (2010), A comparison of two physics-based numerical models for simulating surface water–groundwater interactions, Adv. Water Resour., 33 (4), 456–467, doi:10.1016/j.advwatres.2010.01.010.